# A GRAPH-BASED FRAMEWORK FOR JOINT OOD GENERALIZATION AND DETECTION

## ABSTRACT

In the context of modern machine learning, models deployed in real-world scenarios often encounter various forms of data shifts, leading to challenges in both out-of-distribution (OOD) generalization and detection. While these two aspects have received significant attention individually, they lack a unified framework for theoretical understanding and practical usage. This paper bridges the gap by formalizing a graph-based framework tailored for both OOD generalization and detection. In particular, based on our graph formulation, we introduce spectral learning with wild data (SLW) and show the equivalence of minimizing the objective and performing spectral decomposition on the graph. This equivalence allows us to derive provable error quantifying OOD generalization and detection performance. Empirically, SLW demonstrates competitive performance against existing baselines, aligning with the theoretical insight. Our code is available at https://anonymous.4open.science/r/Anonymous-6074.

## 1 INTRODUCTION

While modern machine learning methods have made substantial strides in recent years, most learning algorithms have been limited to closed-world scenarios, assuming the distribution of training data and labels perfectly aligns with that of the testing data. In reality, models deployed in real-world applications often confront data that deviates from the training distribution in unforeseen ways. As depicted in Figure 1, a model trained on in-distribution (ID) data (e.g., seabirds) may encounter data exhibiting *covariate shifts*, such as birds in forest environments or birds in the cage. In this scenario, the model must retain its ability to accurately classify these covariate-shifted out-of-distribution (OOD) samples as birds—an essential capability known as OOD generalization (Gulrajani & Lopez-Paz, 2021; Koh et al., 2021; Ye et al., 2022). Alternatively, the model may encounter data with novel semantics, like deer or sea lions, which it has not seen during training. In this case, the model must recognize these *semantic-shifted* OOD samples and abstain from making incorrect predictions, underscoring the significance of OOD detection (Yang et al., 2021; Salehi et al., 2022). Thus, for a model to be considered robust and reliable, it must excel in both OOD generalization and detection, tasks that are often addressed separately in current research.

Recently, Bai et al. (2023) introduced a promising direction that addresses both OOD generalization and detection simultaneously. The problem setting leverages unlabeled wild data naturally arising in the model's operational environment, representing it as a composite distribution: $\mathbb{P}_{\text{wild}} := (1 - \pi_s - \pi_c)\mathbb{P}_{\text{in}} + \pi_c\mathbb{P}_{\text{out}}^{\text{covariate}} + \pi_s\mathbb{P}_{\text{out}}^{\text{semantic}}$. Here, $\mathbb{P}_{\text{in}}$, $\mathbb{P}_{\text{out}}^{\text{covariate}}$, and $\mathbb{P}_{\text{out}}^{\text{semantic}}$ represent the marginal distributions of ID, covariate-shifted OOD, and semantic-shifted OOD data, respectively. While such data is ubiquitously available in many real-world applications, harnessing the power of wild data is challenging due to the heterogeneity of the wild data distribution. Moreover, *a formalized understanding of how wild data impacts OOD generalization and detection is still lacking*.

In this paper, we formalize a new graph-based framework tailored for understanding OOD generalization and detection problems jointly. We begin by formulating a graph, where the vertices are all the data points and edges connect similar data points. These edges are defined based on a combination of supervised and self-supervised signals, incorporating both labeled ID data and unlabeled wild data. Importantly, this graph provides a basis for understanding OOD generalization and detection from a spectral analysis perspective, enabling a theoretical characterization of performance through graph factorization. Within this framework, we derive a formal linear probing error, quantifying the

misclassification rate on covariate-shifted OOD data. Furthermore, our framework yields a closed-form solution that quantifies the distance between ID and semantic OOD data, directly impacting OOD detection performance.

Our graph-based framework also illuminates practical algorithm design. Specifically, we present a spectral contrastive loss derived from the spectral decomposition of the graph's adjacency matrix. This loss facilitates joint learning from both labeled ID data and unlabeled wild data, allowing meaningful structures to emerge for OOD generalization and detection (e.g., covariate-shifted OOD data is embedded closely to the ID data, whereas semantic-shifted

| ID | Covariate OOD | Semantic OOD |
|----|---------------|--------------|

Figure 1: Illustration of three types of data in the wild: (1) ID (e.g., seabird), (2) covariate OOD (e.g., bird in the forest and bird in the cage), and (3) semantic OOD (e.g., deer and sea lion).

OOD data is distinguishable from ID data). The algorithm has both practical and theoretical values—(1) it enables end-to-end training in the context of modern neural networks and can be effectively optimized using stochastic gradient descent, making it desirable for real-world applications; (2) it allows drawing a theoretical equivalence between learned representations and spectral decomposition on the graph. Such equivalence facilitates theoretical understanding of the OOD generalization and OOD detection, especially because minimizing the loss is equivalent to performing spectral decomposition on the graph.

Empirical results demonstrate the effectiveness of our learning algorithm, showcasing substantial improvements in both OOD generalization and detection performance. In comparison to the state-of-the-art method SCONE (Bai et al., 2023), our approach achieves a significant reduction in FPR95 by an average of 8.34% across five OOD datasets. We summarize our main contributions below:

1. We propose a novel graph-based framework for understanding both OOD generalization and detection, formalizing it by spectral decomposition of the graph containing ID, covariate-shift OOD data, and semantic-shift OOD data.

2. We provide theoretical insight by analyzing closed-form solutions for the OOD generalization and detection error, based on spectral analysis of the graph.

3. We evaluate our model's performance through a comprehensive set of experiments, providing empirical evidence of its robustness and its alignment with our theoretical analysis. Our model consistently demonstrates strong OOD generalization and OOD detection capabilities, achieving competitive results when benchmarked against the existing state-of-the-art.

## 2 PRELIMINARIES

Bai et al. (2023) proposed to jointly tackle the OOD generalization and OOD detection problems via unlabeled wild data. Inspired by this, we introduce the data setup and learning goal as preliminaries.

**Data setup.** Consider the empirical training set $\mathcal{D}_l \cup \mathcal{D}_u$ as a union of labeled and unlabeled data.

- The labeled set $\mathcal{D}_l = \{\bar{x}_i, y_i\}_{i=1}^n$, where $y_i$ belongs to *known* class space $\mathcal{Y}_l$. Let $\mathbb{P}_{\text{in}}$ denote the marginal distribution over input space, which is referred to as the in-distribution (ID).

- The unlabeled set $\mathcal{D}_u = \{\bar{x}_i\}_{i=1}^m$ consists of ID, covariate OOD, and semantic OOD data, where each sample $\bar{x}_i$ is drawn from the following mixture distribution:

$$\mathbb{P}_{\text{wild}} := (1 - \pi_c - \pi_s)\mathbb{P}_{\text{in}} + \pi_c\mathbb{P}_{\text{out}}^{\text{covariate}} + \pi_s\mathbb{P}_{\text{out}}^{\text{semantic}},$$

where $\pi_c, \pi_s, \pi_c + \pi_s \in [0, 1]$. This mathematical characterization is meaningful since it encapsulates representative distributional shifts that a deployed model may encounter in practice. In particular, $\mathbb{P}_{\text{out}}^{\text{covariate}}$ shares the same label space as the ID data, yet the input space undergoes covariate shift such as styles and domains, which necessitates **OOD generalization** (i.e., predicting the samples correctly into one of the classes in $\mathcal{Y}_l$). On the other hand, $\mathbb{P}_{\text{out}}^{\text{semantic}}$ represents semantically shifted OOD distribution arising from *novel* classes outside $\mathcal{Y}_l$ and therefore should not be predicted by the classification model. Handling $\mathbb{P}_{\text{out}}^{\text{semantic}}$ requires **OOD detection**, which enables obtaining from a prediction.

**Learning goal.** Our learning framework is centered around the construction of two key components, an OOD detector $g_\theta \colon \mathcal{X} \to \{\text{IN}, \text{OUT}\}$ and a multi-class classifier $f_\theta$, by leveraging data from both $\mathbb{P}_{\text{in}}$ and $\mathbb{P}_{\text{wild}}$. Let $\hat{y}(f_\theta(\bar{x})) := \text{argmax}_y f_\theta^{(y)}(\bar{x})$, where $f_\theta^{(y)}(\bar{x})$ denotes the $y$-th element of $f_\theta(\bar{x})$, corresponding to label $y$. We notate $g_\theta$ and $f_\theta$ with parameters $\theta$ to indicate that these functions share neural network parameters. In our model evaluation, we are interested in the following metrics:

$$(\text{ID generalization}) \quad \uparrow \text{ID-Acc}(f_\theta) := \mathbb{E}_{(\bar{x},y)\sim\mathbb{P}_{\text{in}}}(\mathbb{1}\{\hat{y}(f_\theta(\bar{x})) = y\}),$$
$$(\text{OOD generalization}) \quad \uparrow \text{OOD-Acc}(f_\theta) := \mathbb{E}_{(\bar{x},y)\sim\mathbb{P}_{\text{out}}^{\text{covariate}}}(\mathbb{1}\{\hat{y}(f_\theta(\bar{x})) = y\}),$$
$$(\text{OOD detection}) \quad \downarrow \text{FPR}(g_\theta) := \mathbb{E}_{\bar{x}\sim\mathbb{P}_{\text{out}}^{\text{semantic}}}(\mathbb{1}\{g_\theta(\bar{x}) = \text{IN}\}),$$

where $\mathbb{1}\{\cdot\}$ represents the indicator function, while the arrows indicate the directionality of improvement (higher/lower is better). For OOD detection, ID samples are considered positive and FPR signifies the false positive rate.

## 3 METHODOLOGY

In this section, we present a new graph-based framework for tackling both OOD generalization and detection problems. This framework enables us to gain theoretical insight into the learned embedding space by spectral decomposition on a graph, where the vertices are the combination of wild data and the labeled ID data, and edges connect similar data points. In what follows, we first introduce the graph-based formulation (Section 3.1). Then, we introduce a loss that performs spectral decomposition on the graph, which can be reformulated as a contrastive learning objective on neural net representations (Section 3.2).

### 3.1 GRAPH FORMULATION

We use $\bar{x}$ to denote the set of all natural data (raw inputs without augmentation). Given an $\bar{x}$, we use $\mathcal{T}(x|\bar{x})$ to denote the probability of $x$ being augmented from $\bar{x}$, and $\mathcal{T}(\cdot|\bar{x})$ to denote the distribution of its augmentation. For instance, when $\bar{x}$ represents an image, $\mathcal{T}(\cdot|\bar{x})$ can be the distribution of common augmentations (Chen et al., 2020a) such as Gaussian blur, color distortion, and random cropping. We define $\mathcal{X}$ as a general population space, which contains the set of all augmented data. In our case, $\mathcal{X}$ is composed of augmented samples from both labeled data $\mathcal{X}_l$ and unlabeled data $\mathcal{X}_u$, with cardinality $|\mathcal{X}| = N$.

We define the graph $G(\mathcal{X}, w)$ with vertex set $\mathcal{X}$ and edge weights $w$. Given our data setup, edge weights $w$ can be decomposed into two components: (1) *self-supervised connectivity* $w^{(u)}$ by treating all points in $\mathcal{X}$ as entirely unlabeled, and (2) *supervised connectivity* $w^{(l)}$ by incorporating labeled information from $\mathcal{X}_l$ to the graph. We proceed to define these two types of connectivity.

**Edge connectivity.** First, by treating all points as unlabeled, we can define self-supervised connectivity. For any two augmented data $x, x' \in \mathcal{X}$, $w_{xx'}^{(u)}$ denotes the marginal probability of generating the positive pair:

$$w_{xx'}^{(u)} \triangleq \mathbb{E}_{\bar{x}\sim\mathbb{P}}\mathcal{T}(x|\bar{x})\mathcal{T}(x'|\bar{x}), \tag{1}$$

where $x$ and $x'$ are augmented from the same image $\bar{x} \sim \mathbb{P}$, and $\mathbb{P}$ is the marginal distribution of both labeled and unlabeled data.

Different from self-supervised learning (Chen et al., 2020a;b;c), we have access to the labeled information for a subset of nodes, which allows adding additional supervised connectivity to the graph. In particular, we consider $(x, x')$ a positive pair when $x$ and $x'$ are augmented from two labeled samples $\bar{x}_l$ and $\bar{x}_l'$ with the same known class $i \in \mathcal{Y}_l$.

Considering both self-supervised and supervised connectivities, the overall similarity for any pair of data $(x, x')$ is formulated as:

$$w_{xx'} = \eta_u w_{xx'}^{(u)} + \eta_l w_{xx'}^{(l)}, \text{where } w_{xx'}^{(l)} \triangleq \sum_{i\in\mathcal{Y}_l} \mathbb{E}_{\bar{x}_l\sim\mathbb{P}_{l_i}}\mathbb{E}_{\bar{x}_l'\sim\mathbb{P}_{l_i}}\mathcal{T}(x|\bar{x}_l)\mathcal{T}(x'|\bar{x}_l'), \tag{2}$$

where $\mathbb{P}_{l_i}$ is the distribution of labeled samples with class label $i \in \mathcal{Y}_l$, and the coefficients $\eta_u, \eta_l$ modulate the relative importance between the two terms. $w_x = \sum_{x'\in\mathcal{X}} w_{xx'}$ denotes the total edge weights connected to a vertex $x$.

**Adjacency matrix.** Having established the notion of connectivity, we now introduce the adjacency matrix $A \in \mathbb{R}^{N \times N}$ with entries $A_{xx'} = w_{xx'}$. The adjacency matrix can be decomposed into the summation of self-supervised adjacency matrix $A^{(u)}$ and supervised adjacency matrix $A^{(l)}$:

$$A = \eta_u A^{(u)} + \eta_l A^{(l)}. \tag{3}$$

As a standard technique in graph theory (Chung, 1997), we use the *normalized adjacency matrix*:

$$\tilde{A} \triangleq D^{-\frac{1}{2}} A D^{-\frac{1}{2}}, \tag{4}$$

where $D \in \mathbb{R}^{N \times N}$ is a diagonal matrix with $D_{xx} = w_x$. The normalization balances the degree of each node, reducing the influence of vertices with very large degrees. The normalized adjacency matrix defines the probability of $x$ and $x'$ being considered as the positive pair, which helps connect to the representation learning loss as we show next.

## 3.2 SPECTRAL CONTRASTIVE LEARNING WITH WILD DATA

We present a spectral contrastive loss that can be derived from a spectral decomposition of the graph adjacency matrix $\tilde{A}$ defined above. The loss learns feature representation jointly from both labeled ID data and unlabeled wild data, so that meaningful structures emerge for both OOD generalization and detection (e.g., covariate-shifted OOD data is embedded closely to the ID data, whereas semantic-shifted OOD data is distinguishable from ID data). The algorithm has both practical and theoretical values—it (1) enables end-to-end training in the context of modern neural networks and (2) allows drawing a theoretical equivalence between learned representations and the top-$k$ singular vectors of $\tilde{A}$. Such equivalence facilitates theoretical understanding of the OOD generalization and OOD detection capability encoded in $\tilde{A}$. Specifically, we consider low-rank matrix approximation:

$$\min_{F \in \mathbb{R}^{N \times k}} \mathcal{L}_{\mathrm{mf}}(F, A) \triangleq \left\| \tilde{A} - FF^{\top} \right\|_F^2 \tag{5}$$

According to the Eckart–Young–Mirsky theorem (Eckart & Young, 1936), the minimizer of this loss function is $F_k \in \mathbb{R}^{N \times k}$ such that $F_k F_k^{\top}$ contains the top-$k$ components of $\tilde{A}$'s SVD decomposition.

Now, if we view each row $f_x^{\top}$ of $F$ as a scaled version of learned feature embedding $f : \mathcal{X} \mapsto \mathbb{R}^k$, the $\mathcal{L}_{\mathrm{mf}}(F, A)$ can be written as a form of the contrastive learning objective. This connection is formalized in Theorem 3.1 below, with full proof in the Appendix A.

**Theorem 3.1.** *Let $f_x = \sqrt{w_x} f(x)$ for some function $f$. Recall $\eta_u, \eta_l$ are coefficients defined in Eq. 1. Then, the loss function $\mathcal{L}_{\mathrm{mf}}(F, A)$ is equivalent to the following loss function for $f$, which we term **Spectral Learning with Wild Data (SLW)**:*

$$\mathcal{L}_{SLW}(f) \triangleq -2\eta_u \mathcal{L}_1(f) - 2\eta_l \mathcal{L}_2(f) + \eta_u^2 \mathcal{L}_3(f) + 2\eta_u \eta_l \mathcal{L}_4(f) + \eta_l^2 \mathcal{L}_5(f), \tag{6}$$

*where*

$$\mathcal{L}_1(f) = \sum_{i \in \mathcal{Y}_l} \mathbb{E}_{\substack{\bar{x}_l \sim \mathbb{P}_{l_i}, \bar{x}_l' \sim \mathbb{P}_{l_i}, \\ x \sim \mathcal{T}(\cdot | \bar{x}_l), x^+ \sim \mathcal{T}(\cdot | \bar{x}_l')}} \left[ f(x)^{\top} f\left(x^+\right) \right], \mathcal{L}_2(f) = \mathbb{E}_{\substack{\bar{x}_u \sim \mathbb{P}, \\ x \sim \mathcal{T}(\cdot | \bar{x}_u), x^+ \sim \mathcal{T}(\cdot | \bar{x}_u)}} \left[ f(x)^{\top} f\left(x^+\right) \right],$$

$$\mathcal{L}_3(f) = \sum_{i,j \in \mathcal{Y}_l} \mathbb{E}_{\substack{\bar{x}_l \sim \mathbb{P}_{l_i}, \bar{x}_l' \sim \mathbb{P}_{l_j}, \\ x \sim \mathcal{T}(\cdot | \bar{x}_l), x^- \sim \mathcal{T}(\cdot | \bar{x}_l')}} \left[ \left( f(x)^{\top} f\left(x^-\right) \right)^2 \right],$$

$$\mathcal{L}_4(f) = \sum_{i \in \mathcal{Y}_l} \mathbb{E}_{\substack{\bar{x}_l \sim \mathbb{P}_{l_i}, \bar{x}_u \sim \mathbb{P}, \\ x \sim \mathcal{T}(\cdot | \bar{x}_l), x^- \sim \mathcal{T}(\cdot | \bar{x}_u)}} \left[ \left( f(x)^{\top} f\left(x^-\right) \right)^2 \right], \mathcal{L}_5(f) = \mathbb{E}_{\substack{\bar{x}_u \sim \mathbb{P}, \bar{x}_u' \sim \mathbb{P}, \\ x \sim \mathcal{T}(\cdot | \bar{x}_u), x^- \sim \mathcal{T}(\cdot | \bar{x}_u')}} \left[ \left( f(x)^{\top} f\left(x^-\right) \right)^2 \right].$$

**Interpretation of loss.** At a high level, the loss components $\mathcal{L}_1$ and $\mathcal{L}_2$ contribute to pulling the embeddings of positive pairs closer, while $\mathcal{L}_3$, $\mathcal{L}_4$ and $\mathcal{L}_5$ push apart the embeddings of negative pairs. For positive pairs, $\mathcal{L}_1$ samples two random augmentation views from two images sharing the same class label, and $\mathcal{L}_2$ samples two augmentation views from the same image in $\mathcal{X}$. For negative pairs, $\mathcal{L}_3$ samples two augmentation views from two images in $\mathcal{X}_l$ with any class label; $\mathcal{L}_4$ samples two views of one image in $\mathcal{X}_l$ and another one in $\mathcal{X}$; $\mathcal{L}_5$ samples two views from two random images in $\mathcal{X}$. In particular, our loss components on the positive pairs can pull together samples sharing the

same classes, thereby helping OOD generalization. At the same time, our loss components on the negative pairs can help separate semantic OOD data in the embedding space, thus benefiting OOD detection. The loss is inspired by pioneering works on spectral contrastive learning (HaoChen et al., 2021; Shen et al., 2022; Sun et al., 2023), which analyzed problems such as unsupervised learning, unsupervised domain adaptation, and novel category discovery—all of which assume unlabeled data has homogeneous distribution (e.g., either entirely from $\mathbb{P}_{\text{out}}^{\text{covariate}}$ in case of unsupervised domain adaptation, or entirely from $\mathbb{P}_{\text{out}}^{\text{semantic}}$ in case of novel category discovery). However, our paper focuses on the joint problem of OOD generalization and detection, which has fundamentally different data setup and learning goals (*cf.* Section 2). In particular, we are dealing with unlabeled data with heterogeneous mixture distribution, which is more general and challenging than previous works. We are interested in leveraging labeled data to classify some unlabeled data correctly into the known categories while rejecting the remainder of unlabeled data from new categories, which was not considered in the above works. Accordingly, we derive novel theoretical analysis uniquely tailored to our problem setting, which we present next.

## 4    THEORETICAL ANALYSIS

In this section, we present a theoretical analysis based on the spectral loss. Our formal investigation is centered around the understanding of both OOD generalization and detection.

### 4.1    ANALYTIC FORM OF LEARNED REPRESENTATIONS

To obtain the representations, one can train the neural network $f : \mathcal{X} \mapsto \mathbb{R}^k$ using the spectral loss defined in Equation 6. Minimizing the loss yields representation $Z \in \mathbb{R}^{N \times k}$, where each row vector $z_i = f(x_i)^\top$. According to Theorem 3.1, the closed-form solution for the representations is equivalent to performing spectral decomposition of the adjacency matrix. Thus, we have $F_k = \sqrt{D}Z$, where $F_k F_k^\top$ contains the top-$k$ components of $\tilde{A}$'s SVD decomposition and $D$ is the diagonal matrix. We further define the top-$k$ singular vectors of $\tilde{A}$ as $V_k \in \mathbb{R}^{N \times k}$, so we have $F_k = V_k \sqrt{\Sigma_k}$, where $\Sigma_k$ is a diagonal matrix of the top-$k$ singular values of $\tilde{A}$. By equalizing the two forms of $F_k$, the closed-formed solution of the learned feature space is given by $Z = [D]^{-\frac{1}{2}} V_k \sqrt{\Sigma_k}$.

### 4.2    ANALYSIS TARGET

**Linear probing evaluation.** We assess OOD generalization performance based on the linear probing error, which is commonly used in self-supervised learning (Chen et al., 2020a). Specifically, the weight of a linear classifier is denoted as $\mathbf{M} \in \mathbb{R}^{k \times |\mathcal{Y}_l|}$, which is learned with ID data to minimize the error. The class prediction for an input $\bar{x}$ is given by $h(\bar{x}; f, \mathbf{M}) = \text{argmax}_{i \in \mathcal{Y}_l}(f(\bar{x})^\top \mathbf{M})_i$. The linear probing error measures the misclassification of linear head on covariate-shifted OOD data:

$$\mathcal{E}(f) \triangleq \mathbb{E}_{\bar{x} \sim \mathbb{P}_{\text{out}}^{\text{covariate}}} \mathbb{1}[y(\bar{x}) \neq h(\bar{x}; f, \mathbf{M})], \tag{7}$$

where $y(\bar{x})$ indicates the ground-truth class of $\bar{x}$. $\mathcal{E}(f) = 0$ indicates perfect OOD generalization.

**Separability evaluation.** Based on the closed-form embeddings, we can also quantify the distance between the ID and semantic OOD data:

$$\mathcal{S}(f) \triangleq \mathbb{E}_{\bar{x}_i \sim \mathbb{P}_{\text{in}}, \bar{x}_j \sim \mathbb{P}_{\text{out}}^{\text{semantic}}} \|f(\bar{x}_i) - f(\bar{x}_j)\|_2^2. \tag{8}$$

The magnitude of $\mathcal{S}(f)$ reflects the extent of separation between ID and semantic OOD data. Larger $\mathcal{S}(f)$ suggests better OOD detection capability.

### 4.3    AN ILLUSTRATIVE EXAMPLE

**Setup.** We use an illustrative example to explain our theoretical insights. In Figure 2, the training examples come from 5 types of data: angel in sketch (ID), tiger in sketch (ID), angel in painting (covariate OOD), tiger in painting (covariate OOD), and panda (semantic OOD). The label space $\mathcal{Y}_l$ consists of two known classes: angel and tiger. Class Panda is considered a novel class. The goal is to classify between images of angels and tigers while rejecting images of pandas.

**Augmentation Transformation Probability.** Based on the data setup, we formally define the augmentation transformation, which encodes the probability of augmenting an original image $\bar{x}$ to the augmented view $x$:

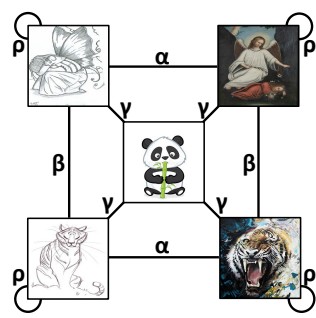

$$\mathcal{T}(x \mid \bar{x}) = \begin{cases} \rho & \text{if} & y(\bar{x}) = y(x), d(\bar{x}) = d(x); \\ \alpha & \text{if} & y(\bar{x}) = y(x), d(\bar{x}) \neq d(x); \\ \beta & \text{if} & y(\bar{x}) \neq y(x), d(\bar{x}) = d(x); \\ \gamma & \text{if} & y(\bar{x}) \neq y(x), d(\bar{x}) \neq d(x). \end{cases} \quad (9)$$

Here $d(\bar{x})$ is the domain of sample $\bar{x}$, and $y(\bar{x})$ is the class label of sample $\bar{x}$. $\alpha$ indicates the augmentation probability when two samples share the same label but different domains, and $\beta$ indicates the probability when two samples share different class labels but with the same domain. It is natural to assume the magnitude order that follows $\rho \gg \max(\alpha, \beta) \geq \min(\alpha, \beta) \gg \gamma \geq 0$.

Figure 2: Illustration of the graph and the augmentation probability.

**Adjacency matrix.** With Eq. 9 and the definition in Section 3.1, we can derive the analytic form of adjacency matrix $A$.

$$\eta_u A^{(u)} = \begin{bmatrix} \rho^2 + \beta^2 + \alpha^2 + 2\gamma^2 & 2\rho\beta + \gamma^2 + 2\gamma\alpha & 2\rho\alpha + \gamma^2 + 2\gamma\beta & 2\alpha\beta + \gamma^2 + 2\gamma\rho & \gamma(\gamma + \alpha + \beta + 2\rho) \\ 2\rho\beta + \gamma^2 + 2\gamma\alpha & \rho^2 + \beta^2 + \alpha^2 + 2\gamma^2 & 2\alpha\beta + \gamma^2 + 2\gamma\rho & 2\rho\alpha + \gamma^2 + 2\gamma\beta & \gamma(\gamma + \alpha + \beta + 2\rho) \\ 2\rho\alpha + \gamma^2 + 2\gamma\beta & 2\alpha\beta + \gamma^2 + 2\gamma\rho & \rho^2 + \beta^2 + \alpha^2 + 2\gamma^2 & 2\rho\beta + \gamma^2 + 2\gamma\alpha & \gamma(\gamma + \alpha + \beta + 2\rho) \\ 2\alpha\beta + \gamma^2 + 2\gamma\rho & 2\rho\alpha + \gamma^2 + 2\gamma\beta & 2\rho\beta + \gamma^2 + 2\gamma\alpha & \rho^2 + \beta^2 + \alpha^2 + 2\gamma^2 & \gamma(\gamma + \alpha + \beta + 2\rho) \\ \gamma(\gamma + \alpha + \beta + 2\rho) & \gamma(\gamma + \alpha + \beta + 2\rho) & \gamma(\gamma + \alpha + \beta + 2\rho) & \gamma(\gamma + \alpha + \beta + 2\rho) & \rho^2 + 4\gamma^2 \end{bmatrix} \quad (10)$$

$$A = \frac{1}{C}(\eta_l A^{(l)} + \eta_u A^{(u)}) = \frac{1}{C}\left(\begin{bmatrix} \rho^2 + \beta^2 & 2\rho\beta & \rho\alpha + \gamma\beta & \alpha\beta + \gamma\rho & \gamma(\rho + \beta) \\ 2\rho\beta & \rho^2 + \beta^2 & \alpha\beta + \gamma\rho & \rho\alpha + \gamma\beta & \gamma(\rho + \beta) \\ \rho\alpha + \gamma\beta & \alpha\beta + \gamma\rho & \alpha^2 + \gamma^2 & 2\gamma\alpha & \gamma(\alpha + \gamma) \\ \alpha\beta + \gamma\rho & \rho\alpha + \gamma\beta & 2\gamma\alpha & \alpha^2 + \gamma^2 & \gamma(\alpha + \gamma) \\ \gamma(\rho + \beta) & \gamma(\rho + \beta) & \gamma(\alpha + \gamma) & \gamma(\alpha + \gamma) & 2\gamma^2 \end{bmatrix} + \eta_u A^{(u)}\right), \quad (11)$$

where $C$ is the normalization constant to ensure the summation of weights amounts to 1. Each row or column encodes connectivity associated with a specific sample, ordered by: angel sketch, tiger sketch, angel painting, tiger painting, and panda. We refer readers to Appendix D.1 for the detailed derivation.

**Main analysis.** We are primarily interested in analyzing the representation space derived from $A$. We mainly put analysis on the top-3 eigenvectors $\widehat{V} \in \mathbb{R}^{5 \times 3}$ and measure both the linear probing error and separability.

**Theorem 4.1.** *Assume $\eta_u = 5, \eta_l = 1$, we have:*

$$\widehat{V} = \begin{cases} \begin{bmatrix} \frac{1}{\sqrt{3}} & \frac{1}{\sqrt{3}} & \frac{1}{\sqrt{6}} & \frac{1}{\sqrt{6}} & 0 \\ 0 & 0 & 0 & 0 & 1 \\ -\frac{1}{\sqrt{3}} & \frac{1}{\sqrt{3}} & -\frac{1}{\sqrt{6}} & \frac{1}{\sqrt{6}} & 0 \end{bmatrix}^{\top} & , \text{if } \frac{9}{8}\alpha > \beta; \\ \begin{bmatrix} \frac{1}{\sqrt{3}} & \frac{1}{\sqrt{3}} & \frac{1}{\sqrt{6}} & \frac{1}{\sqrt{6}} & 0 \\ 0 & 0 & 0 & 0 & 1 \\ -\frac{1}{\sqrt{6}} & -\frac{1}{\sqrt{6}} & \frac{1}{\sqrt{3}} & \frac{1}{\sqrt{3}} & 0 \end{bmatrix}^{\top} & , \text{if } \frac{9}{8}\alpha < \beta. \end{cases} \quad , \mathcal{E}(f) = \begin{cases} 0 & , \text{if } \frac{9}{8}\alpha > \beta; \\ 2 & , \text{if } \frac{9}{8}\alpha < \beta. \end{cases} \quad (12)$$

**Interpretation.** The discussion can be divided into two cases: (1) $\frac{9}{8}\alpha > \beta$. (2) $\frac{9}{8}\alpha < \beta$. In the first case when the connection between the class (multiplied by $\frac{9}{8}$) is stronger than the domain, the model could learn a perfect ID classifier based on features in the first two rows in $V$ and effectively generalize to the covariate-shifted domain (the third and fourth row in $\widehat{V}$), achieving perfect OOD generalization with linear probing error $\mathcal{E}(f) = 0$. In the second case when the connection between the domain is stronger than the connection between the class (scaled by $\frac{9}{8}$), the embeddings of covariate-shifted OOD data are identical, resulting in high OOD generalization error.

**Theorem 4.2.** *Denote $\alpha' = \frac{\alpha}{\rho}$ and $\beta' = \frac{\beta}{\rho}$ and assume $\eta_u = 5, \eta_l = 1$, we have:*

$$\mathcal{S}(f) = \begin{cases} (7 + 12\beta' + 12\alpha')(\frac{1 - 2\beta'}{3}(1 - \beta' - \frac{3}{4}\alpha')^2 + 1) & , \text{if } \frac{9}{8}\alpha > \beta; \\ (7 + 12\beta' + 12\alpha')(\frac{2 - 3\alpha'}{8}(1 - \beta' - \frac{3}{4}\alpha')^2 + 1) & , \text{if } \frac{9}{8}\alpha < \beta. \end{cases} \quad (13)$$

**Interpretation.** We analyze the function $S(f)$ under different $\alpha'$ and $\beta'$ values in Figure 3. Overall the distance between semantic OOD data and ID data displays a large value, which facilitates OOD detection. Note that a clear boundary in Figure 3 indicates $\frac{9}{8}\alpha = \beta$.

**More analysis.** Building upon the understanding of both OOD generalization and detection, we further discuss the influence of different semantic OOD data in Appendix B, and the impact of ID labels in Appendix C.

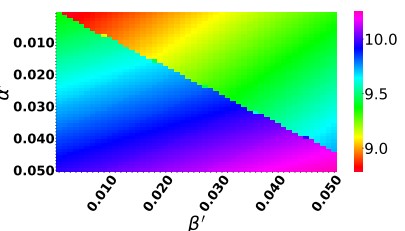

Figure 3: Value of function $S(f)$

## 5 EXPERIMENTS

Beyond theoretical insights, we show empirically that SLW is competitive. We present the experimental setup in Section 5.1, results in Section 5.2, and further analysis in Section 5.3.

### 5.1 EXPERIMENTAL SETUP

**Datasets and benchmarks.** Following the setup of Bai et al. (2023), we employ CIFAR-10 (Krizhevsky et al., 2009) as $\mathbb{P}_{in}$ and CIFAR-10-C (Hendrycks & Dietterich, 2018) with Gaussian additive noise as the $\mathbb{P}_{out}^{covariate}$. For $\mathbb{P}_{out}^{semantic}$, we leverage SVHN (Netzer et al., 2011), LSUN (Yu et al., 2015), Places365 (Zhou et al., 2017), Textures (Cimpoi et al., 2014). To simulate the wild distribution $\mathbb{P}_{wild}$, we adopt the same mixture ratio as in SCONE (Bai et al., 2023), where $\pi_c = 0.5$ and $\pi_s = 0.1$. Detailed descriptions of the datasets and data mixture can be found in the Appendix E.1. Large-scale results on the ImageNet dataset can be found in Appendix E.2. Additional results on the Office-Home (Venkateswara et al., 2017) can be found in Appendix E.3.

**Implementation details.** We adopt Wide ResNet (Zagoruyko & Komodakis, 2016) with 40 layers and a widen factor of 2. We use stochastic gradient descent with Nesterov momentum (Duchi et al., 2011), with weight decay 0.0005 and momentum 0.09. We divide CIFAR-10 training set into 50% labeled as ID and 50% unlabeled. And we mix unlabeled CIFAR-10, CIFAR-10-C, and semantic OOD data to generate the wild dataset. Starting from random initialization, we train the network with the loss function in Eq. 6 for 1000 epochs. The learning rate is 0.03 and the batch size is 512. $\eta_u$ is selected within $\{1.00, 2.00\}$ and $\eta_l$ is within $\{0.02, 0.10, 0.50, 1.00\}$. Subsequently, we follow the standard approach (Shen et al., 2022) and use labeled ID data to fine-tune the model with cross-entropy loss for better generalization ability. We fine-tune for 20 epochs with a learning rate of 0.005 and batch size of 512. The fine-tuned model is used to evaluate the OOD generalization and OOD detection performance. We utilize a distance-based method for OOD detection, which resonates with our theoretical analysis. Specifically, our default approach employs a simple non-parametric KNN distance (Sun et al., 2022), which does not impose any distributional assumption on the feature space. The threshold is determined based on the clean ID set at 95% percentile. For further implementation details, hyper-parameters, and validation strategy, please see Appendix F.

### 5.2 RESULTS AND DISCUSSION

**Competitive empirical performance.** The main results in Table 1 demonstrate that our method not only enjoys theoretical guarantees but also exhibits competitive empirical performance compared to existing baselines. For a comprehensive evaluation, we consider three groups of methods for OOD generalization and OOD detection. Closest to our setting, we compare with strong baselines trained with wild data, namely OE (Hendrycks et al., 2018), Energy-regularized learning (Liu et al., 2020), Woods (Katz-Samuels et al., 2022), and Scone (Bai et al., 2023).

The empirical results provide interesting insights into the performance of various methods for OOD detection and generalization. **(1)** Methods tailored for OOD detection tend to capture the domain-variant information and struggle with the covariate distribution shift, resulting in suboptimal OOD accuracy. **(2)** While approaches for OOD generalization demonstrate improved OOD accuracy, they cannot effectively distinguish between ID data and semantic OOD data, leading to poor OOD detection performance. **(3)** Methods trained with wild data emerge as robust OOD detectors, yet display a notable decline in OOD generalization, highlighting the confusion introduced by covariate OOD data. In contrast, our method excels in both OOD detection and generalization performance. Our method even surpasses the latest method SCONE by **25.10%** in terms of FPR95 on the Textures

| Method | SVHN $\mathbb{P}_{out}^{semantic}$, CIFAR-10-C $\mathbb{P}_{out}^{covariate}$ | | | | LSUN-C $\mathbb{P}_{out}^{semantic}$, CIFAR-10-C $\mathbb{P}_{out}^{covariate}$ | | | | Textures $\mathbb{P}_{out}^{semantic}$, CIFAR-10-C $\mathbb{P}_{out}^{covariate}$ | | | |
|---|---|---|---|---|---|---|---|---|---|---|---|---|
| | OOD Acc.↑ | ID Acc.↑ | FPR↓ | AUROC↑ | OOD Acc.↑ | ID Acc.↑ | FPR↓ | AUROC↑ | OOD Acc.↑ | ID Acc.↑ | FPR↓ | AUROC↑ |
| *OOD detection* | | | | | | | | | | | | |
| **MSP** | 75.05 | 94.84 | 48.49 | 91.89 | 75.05 | 94.84 | 30.80 | 95.65 | 75.05 | 94.84 | 59.28 | 88.50 |
| **ODIN** | 75.05 | 94.84 | 33.35 | 91.96 | 75.05 | 94.84 | 15.52 | 97.04 | 75.05 | 94.84 | 49.12 | 84.97 |
| **Energy** | 75.05 | 94.84 | 35.59 | 90.96 | 75.05 | 94.84 | 8.26 | 98.35 | 75.05 | 94.84 | 52.79 | 85.22 |
| **Mahalanobis** | 75.05 | 94.84 | 12.89 | 97.62 | 75.05 | 94.84 | 39.22 | 94.15 | 75.05 | 94.84 | 15.00 | 97.33 |
| **ViM** | 75.05 | 94.84 | 21.95 | 95.48 | 75.05 | 94.84 | 5.90 | 98.82 | 75.05 | 94.84 | 29.35 | 93.70 |
| **KNN** | 75.05 | 94.84 | 28.92 | 95.71 | 75.05 | 94.84 | 28.08 | 95.33 | 75.05 | 94.84 | 39.50 | 92.73 |
| **ASH** | 75.05 | 94.84 | 40.76 | 90.16 | 75.05 | 94.84 | 2.39 | 99.35 | 75.05 | 94.84 | 53.37 | 85.63 |
| *OOD generalization* | | | | | | | | | | | | |
| **IRM** | 77.92 | 90.85 | 63.65 | 90.70 | 77.92 | 90.85 | 36.67 | 94.22 | 77.92 | 90.85 | 59.42 | 87.81 |
| **GroupDRO** | 77.27 | **94.97** | 23.78 | 94.93 | 77.27 | 94.97 | 6.90 | 98.51 | 77.27 | **94.97** | 62.08 | 84.60 |
| **Mixup** | 79.17 | 93.30 | 97.33 | 18.78 | 79.17 | 93.30 | 52.10 | 76.66 | 79.17 | 93.30 | 58.24 | 75.70 |
| **VREx** | 76.90 | 91.35 | 55.92 | 91.22 | 76.90 | 91.35 | 51.50 | 91.56 | 76.90 | 91.35 | 65.45 | 85.46 |
| **EQRM** | 75.71 | 92.93 | 51.86 | 90.92 | 75.71 | 92.93 | 21.53 | 96.49 | 75.71 | 92.93 | 57.18 | 89.11 |
| SharpDRO | 79.03 | 94.91 | 21.24 | 96.14 | 79.03 | 94.91 | 5.67 | 98.71 | 79.03 | 94.91 | 42.94 | 89.99 |
| *Learning w. $\mathbb{P}_{wild}$* | | | | | | | | | | | | |
| **OE** | 37.61 | 94.68 | 0.84 | 99.80 | 41.37 | 93.99 | 3.07 | 99.26 | 44.71 | 92.84 | 29.36 | 93.93 |
| **Energy (w. outlier)** | 20.74 | 90.22 | 0.86 | 99.81 | 32.55 | 92.97 | 2.33 | **99.93** | 49.34 | 94.68 | 16.42 | 96.46 |
| **Woods** | 52.76 | 94.86 | 2.11 | 99.52 | 76.90 | **95.02** | 1.80 | 99.56 | 83.14 | 94.49 | 39.10 | 90.45 |
| **Scone** | 84.69 | 94.65 | 10.86 | 97.84 | 84.58 | 93.73 | 10.23 | 98.02 | **85.56** | 93.97 | 37.15 | 90.91 |
| **SLW (Ours)** | **86.62**$_{\pm0.3}$ | 93.10$_{\pm0.1}$ | **0.13**$_{\pm0.0}$ | **99.98**$_{\pm0.0}$ | **85.88**$_{\pm0.2}$ | 92.61$_{\pm0.1}$ | **1.76**$_{\pm0.8}$ | 99.75$_{\pm0.1}$ | 81.40$_{\pm0.7}$ | 92.50$_{\pm0.1}$ | **12.05**$_{\pm0.8}$ | **98.25**$_{\pm0.2}$ |

Table 1: Main results: comparison with competitive OOD generalization and OOD detection methods on CIFAR-10. Additional results for the Places365 and LSUN-R datasets can be found in Table 2. **Bold**=best. (*Since all the OOD detection methods use the same model trained with the CE loss on $\mathbb{P}_{in}$, they display the same ID and OOD accuracy on CIFAR-10-C.)

dataset. Methodologically, SCONE uses constrained optimization whereas our method brings a novel graph-based perspective. More results can be found in the Appendix E.

## 5.3 FURTHER ANALYSIS

**Visualization of OOD detection score distributions.** In Figure 4 (a), we visualize the distribution of KNN distances. The KNN scores are computed based on samples from the test set after contrastive training and fine-tuning stages. There are two salient observations: First, our learning framework effectively pushes the semantic OOD data (in blue) to be apart from the ID data (in orange) in the embedding space, which benefits OOD detection. Moreover, as evidenced by the small KNN distance, covariate-shifted OOD data (in green) is embedded closely to the ID data, which aligns with our expectations.

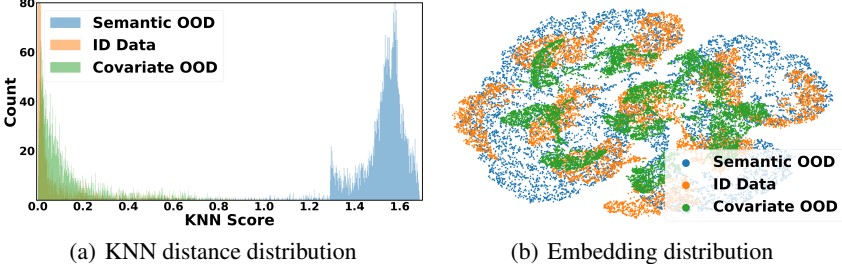

(a) KNN distance distribution                (b) Embedding distribution

Figure 4: (a) Distribution of KNN distance. (b) t-SNE visualization of learned embeddings. We employ CIFAR-10 as $\mathbb{P}_{in}$, CIFAR-10-C as $\mathbb{P}_{out}^{covariate}$, and SVHN as $\mathbb{P}_{out}^{semantic}$.

**Visualization of embeddings.** Figure 4 (b) displays the t-SNE visualization (Van der Maaten & Hinton, 2008) visualization of the normalized penultimate-layer embeddings. Samples are from the test set of ID, covariate OOD, and semantic OOD data, respectively. The visualization demonstrates the alignment of ID and covariate OOD data in the embedding space, which allows the classifier learned on the ID data to extrapolate to the covariate OOD data and thereby benefiting OOD generalization.

## 6 RELATED WORKS

**Out-of-distribution detection.** OOD detection has gained soaring research attention in recent years. The current research track can be divided into post hoc and regularization-based methods. Post hoc methods derive OOD scores at test-time based on a pre-trained model, which can be categorized as confidence-based methods (Bendale & Boult, 2016; Hendrycks & Gimpel, 2017; Liang et al., 2018), energy-based methods (Liu et al., 2020; Wang et al., 2021a; Sun et al., 2021; Sun & Li, 2022; Morteza & Li, 2022; Djurisic et al., 2023), distance-based methods (Lee et al., 2018; Zhou et al., 2021b; Sehwag et al., 2021; Sun et al., 2022; Du et al., 2022; Ming et al., 2022a; 2023), and gradient-based method (Huang et al., 2021). On the other hand, regularization-based methods

aim to train the OOD detector by training-time regularization. Most approaches require auxiliary OOD data (Bevandic et al., 2018; Geifman & El-Yaniv, 2019; Mohseni et al., 2020; Hendrycks et al., 2018; Ming et al., 2022b). However, a limitation of existing methods is the reliance on clean semantic OOD datasets for training. To address this challenge, Katz-Samuels et al. (2022) first explored the use of wild data, which includes unlabeled ID and semantic OOD data. Building upon this idea, Bai et al. (2023) extended the characterization of wild data to encompass ID, covariate OOD, and semantic OOD data, providing a more generalized data mixture in practice. In our paper, we provide a novel graph-based approach for understanding both OOD generalization and detection based on the setup proposed by Bai et al. (2023).

**Out-of-distribution generalization.** OOD generalization aims to learn domain-invariant representations that can effectively generalize to unseen domains, which is more challenging than classic domain adaptation problem (Ganin & Lempitsky, 2015; Chen et al., 2018; Zhang et al., 2019; Cui et al., 2020), where the model has access to unlabeled data from the target domain. OOD generalization and domain generalization (Wang et al., 2023) focus on capturing semantic features that remain consistent across diverse domains, which can be categorized as reducing feature discrepancies across the source domains (Li et al., 2018b;c; Arjovsky et al., 2019; Zhao et al., 2020; Ahuja et al., 2021), ensemble and meta learning (Balaji et al., 2018; Li et al., 2018a; 2019; Zhang et al., 2021; Bui et al., 2021), robust optimization (Cha et al., 2021; Krueger et al., 2021; Sagawa et al., 2020; Shi et al., 2022; Ramé et al., 2022), augmentation (Zhou et al., 2020; Nam et al., 2021; Nuriel et al., 2021; Zhou et al., 2021a), and disentanglement (Zhang et al., 2022). Distinct from prior literature about generalization, Bai et al. (2023) introduces a framework that leverages the wild data ubiquitous in the real world, aiming to build a robust classifier and a reliable OOD detector simultaneously. Based on the problem setting introduced by Bai et al. (2023), we contribute novel theoretical insights into the understanding of both OOD generalization and detection.

**Spectral graph theory.** Spectral graph theory is a classic research field (Chung, 1997; McSherry, 2001; Kannan et al., 2004; Lee et al., 2014; Cheeger, 2015), concerning the study of graph partitioning through analyzing the eigenspace of the adjacency matrix. The spectral graph theory is also widely applied in machine learning Shi & Malik (2000); Blum (2001); Ng et al. (2001); Zhu et al. (2003); Argyriou et al. (2005); Shaham et al. (2018). Recently, HaoChen et al. (2021) presented unsupervised spectral contrastive loss derived from the factorization of the graph's adjacency matrix. Shen et al. (2022) provided a graph-theoretical analysis for unsupervised domain adaptation based on the assumption of unlabeled data entirely from $\mathbb{P}_{out}^{covariate}$. Sun et al. (2023) first introduced the label information and explored novel category discovery, considering unlabeled data covers $\mathbb{P}_{out}^{semantic}$. All of the previous literature assumed unlabeled data has a homogeneous distribution. In contrast, our work focuses on the joint problem of OOD generalization and detection, tackling the challenge of unlabeled data characterized by a heterogeneous mixture distribution, which is a more general and complex scenario than previous works.

**Contrastive learning.** Recent works on contrastive learning advance the development of deep neural networks with a huge empirical success (Chen et al., 2020a;b;c; Grill et al., 2020; Hu et al., 2021; Caron et al., 2020; Chen & He, 2021; Bardes et al., 2022; Zbontar et al., 2021). Simultaneously, many theoretical works establish the foundation for understanding representations learned by contrastive learning through linear probing evaluation (Saunshi et al., 2019; Lee et al., 2021; Tosh et al., 2021a;b; Balestriero & LeCun, 2022; Shi et al., 2023). HaoChen et al. (2021; 2022); Sun et al. (2023) extended the understanding and providing error analyses for different downstream tasks. Orthogonal to prior works, we provide a graph-based framework tailored for the wild environment to understand both OOD generalization and detection.

## 7 CONCLUSION

In this paper, we present a new graph-based framework to jointly tackle both OOD generalization and detection problems. Specifically, we learn representations through Spectral Learning with Wild Data (SLW). The equivalence of minimizing the loss and factorizing the graph's adjacency matrix allows us to draw theoretical insight into both OOD generalization and detection performance. We analyze the closed-form solutions of linear probing error for OOD generalization, as well as separability quantifying OOD detection capability via the distance between the ID and semantic OOD data. Empirically, our framework demonstrates competitive performance against existing baselines, closely aligning with our theoretical insights. We believe our theoretical framework and findings will inspire the community to further union and understand both OOD generalization and detection.

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

## A  Technical Details of Spectral Learning with Wild Data

**Theorem A.1.** *(Recap of Theorem 3.1) Let $f_x = \sqrt{w_x} f(x)$ for some function $f$. Recall $\eta_u, \eta_l$ are coefficients defined in Eq. 1. Then, the loss function $\mathcal{L}_{\mathrm{mf}}(F, A)$ is equivalent to the following loss function for $f$, which we term **Spectral Learning with Wild Data (SLW)**:*

$$\mathcal{L}_{SLW}(f) \triangleq -2\eta_u \mathcal{L}_1(f) - 2\eta_l \mathcal{L}_2(f) + \eta_u^2 \mathcal{L}_3(f) + 2\eta_u \eta_l \mathcal{L}_4(f) + \eta_l^2 \mathcal{L}_5(f), \tag{14}$$

*where*

$$\mathcal{L}_1(f) = \sum_{i \in \mathcal{Y}_l} \mathbb{E}_{\substack{\bar{x}_l \sim \mathbb{P}_{l_i}, \bar{x}'_l \sim \mathbb{P}_{l_i}, \\ x \sim \mathcal{T}(\cdot|\bar{x}_l), x^+ \sim \mathcal{T}(\cdot|\bar{x}'_l)}} \left[ f(x)^\top f\left(x^+\right) \right],$$

$$\mathcal{L}_2(f) = \mathbb{E}_{\substack{\bar{x}_u \sim \mathbb{P}, \\ x \sim \mathcal{T}(\cdot|\bar{x}_u), x^+ \sim \mathcal{T}(\cdot|\bar{x}_u)}} \left[ f(x)^\top f\left(x^+\right) \right],$$

$$\mathcal{L}_3(f) = \sum_{i,j \in \mathcal{Y}_l} \mathbb{E}_{\substack{\bar{x}_l \sim \mathbb{P}_{l_i}, \bar{x}'_l \sim \mathbb{P}_{l_j}, \\ x \sim \mathcal{T}(\cdot|\bar{x}_l), x^- \sim \mathcal{T}(\cdot|\bar{x}'_l)}} \left[ \left( f(x)^\top f\left(x^-\right) \right)^2 \right],$$

$$\mathcal{L}_4(f) = \sum_{i \in \mathcal{Y}_l} \mathbb{E}_{\substack{\bar{x}_l \sim \mathbb{P}_{l_i}, \bar{x}_u \sim \mathbb{P}, \\ x \sim \mathcal{T}(\cdot|\bar{x}_l), x^- \sim \mathcal{T}(\cdot|\bar{x}_u)}} \left[ \left( f(x)^\top f\left(x^-\right) \right)^2 \right],$$

$$\mathcal{L}_5(f) = \mathbb{E}_{\substack{\bar{x}_u \sim \mathbb{P}, \bar{x}'_u \sim \mathbb{P}, \\ x \sim \mathcal{T}(\cdot|\bar{x}_u), x^- \sim \mathcal{T}(\cdot|\bar{x}'_u)}} \left[ \left( f(x)^\top f\left(x^-\right) \right)^2 \right].$$

*Proof.* We can expand $\mathcal{L}_{\mathrm{mf}}(F, A)$ and obtain

$$\mathcal{L}_{\mathrm{mf}}(F, A) = \sum_{x,x' \in \mathcal{X}} \left( \frac{w_{xx'}}{\sqrt{w_x w_{x'}}} - f_x^\top f_{x'} \right)^2$$

$$= \text{const} + \sum_{x,x' \in \mathcal{X}} \left( -2 w_{xx'} f(x)^\top f\left(x'\right) + w_x w_{x'} \left( f(x)^\top f\left(x'\right) \right)^2 \right),$$

where $f_x = \sqrt{w_x} f(x)$ is a re-scaled version of $f(x)$. At a high level, we follow the proof in HaoChen et al. (2021), while the specific form of loss varies with the different definitions of positive/negative pairs. The form of $\mathcal{L}_{\mathrm{SLW}}(f)$ is derived from plugging $w_{xx'}$ and $w_x$.

Recall that $w_{xx'}$ is defined by

$$w_{xx'} = \eta_u \sum_{i \in \mathcal{Y}_l} \mathbb{E}_{\bar{x}_l \sim \mathbb{P}_{l_i}} \mathbb{E}_{\bar{x}'_l \sim \mathbb{P}_{l_i}} \mathcal{T}(x|\bar{x}_l) \mathcal{T}\left(x'|\bar{x}'_l\right) + \eta_l \mathbb{E}_{\bar{x}_u \sim \mathbb{P}} \mathcal{T}(x|\bar{x}_u) \mathcal{T}\left(x'|\bar{x}_u\right),$$

and $w_x$ is given by

$$w_x = \sum_{x'} w_{xx'}$$

$$= \eta_u \sum_{i \in \mathcal{Y}_l} \mathbb{E}_{\bar{x}_l \sim \mathbb{P}_{l_i}} \mathbb{E}_{\bar{x}'_l \sim \mathbb{P}_{l_i}} \mathcal{T}(x|\bar{x}_l) \sum_{x'} \mathcal{T}\left(x'|\bar{x}'_l\right) + \eta_l \mathbb{E}_{\bar{x}_u \sim \mathbb{P}} \mathcal{T}(x|\bar{x}_u) \sum_{x'} \mathcal{T}\left(x'|\bar{x}_u\right)$$

$$= \eta_u \sum_{i \in \mathcal{Y}_l} \mathbb{E}_{\bar{x}_l \sim \mathbb{P}_{l_i}} \mathcal{T}(x|\bar{x}_l) + \eta_l \mathbb{E}_{\bar{x}_u \sim \mathbb{P}} \mathcal{T}(x|\bar{x}_u).$$

Plugging in $w_{xx'}$ we have,

$$-2\sum_{x,x'\in\mathcal{X}}w_{xx'}f(x)^\top f(x')$$

$$=-2\sum_{x,x^+\in\mathcal{X}}w_{xx^+}f(x)^\top f\left(x^+\right)$$

$$=-2\eta_u\sum_{i\in\mathcal{Y}_l}\mathbb{E}_{\bar{x}_l\sim\mathbb{P}_{l_i}}\mathbb{E}_{\bar{x}'_l\sim\mathbb{P}_{l_i}}\sum_{x,x'\in\mathcal{X}}\mathcal{T}(x|\bar{x}_l)\mathcal{T}(x'|\bar{x}'_l)f(x)^\top f(x')$$

$$-2\eta_l\mathbb{E}_{\bar{x}_u\sim\mathbb{P}}\sum_{x,x'}\mathcal{T}(x|\bar{x}_u)\mathcal{T}(x'|\bar{x}_u)f(x)^\top f(x')$$

$$=-2\eta_u\sum_{i\in\mathcal{Y}_l}\underset{\substack{\bar{x}_l\sim\mathbb{P}_{l_i},\bar{x}'_l\sim\mathbb{P}_{l_i},\\x\sim\mathcal{T}(\cdot|\bar{x}_l),x^+\sim\mathcal{T}(\cdot|\bar{x}'_l)}}{\mathbb{E}}\left[f(x)^\top f\left(x^+\right)\right]$$

$$-2\eta_l\underset{\substack{\bar{x}_u\sim\mathbb{P},\\x\sim\mathcal{T}(\cdot|\bar{x}_u),x^+\sim\mathcal{T}(\cdot|\bar{x}_u)}}{\mathbb{E}}\left[f(x)^\top f\left(x^+\right)\right]$$

$$=-2\eta_u\mathcal{L}_1(f)-2\eta_l\mathcal{L}_2(f).$$

Plugging $w_x$ and $w_{x'}$ we have,

$$\sum_{x,x'\in\mathcal{X}}w_xw_{x'}\left(f(x)^\top f\left(x'\right)\right)^2$$

$$=\sum_{x,x^-\in\mathcal{X}}w_xw_{x^-}\left(f(x)^\top f\left(x^-\right)\right)^2$$

$$=\sum_{x,x'\in\mathcal{X}}\left(\eta_u\sum_{i\in\mathcal{Y}_l}\mathbb{E}_{\bar{x}_l\sim\mathbb{P}_{l_i}}\mathcal{T}(x|\bar{x}_l)+\eta_l\mathbb{E}_{\bar{x}_u\sim\mathbb{P}}\mathcal{T}(x|\bar{x}_u)\right)$$

$$\cdot\left(\eta_u\sum_{j\in\mathcal{Y}_l}\mathbb{E}_{\bar{x}'_l\sim\mathbb{P}_{l_j}}\mathcal{T}(x^-|\bar{x}'_l)+\eta_l\mathbb{E}_{\bar{x}'_u\sim\mathbb{P}}\mathcal{T}(x^-|\bar{x}'_u)\right)\left(f(x)^\top f\left(x^-\right)\right)^2$$

$$=\eta_u^2\sum_{x,x^-\in\mathcal{X}}\sum_{i\in\mathcal{Y}_l}\mathbb{E}_{\bar{x}_l\sim\mathbb{P}_{l_i}}\mathcal{T}(x|\bar{x}_l)\sum_{j\in\mathcal{Y}_l}\mathbb{E}_{\bar{x}'_l\sim\mathbb{P}_{l_j}}\mathcal{T}(x^-|\bar{x}'_l)\left(f(x)^\top f\left(x^-\right)\right)^2$$

$$+2\eta_u\eta_l\sum_{x,x^-\in\mathcal{X}}\sum_{i\in\mathcal{Y}_l}\mathbb{E}_{\bar{x}_l\sim\mathbb{P}_{l_i}}\mathcal{T}(x|\bar{x}_l)\mathbb{E}_{\bar{x}_u\sim\mathbb{P}}\mathcal{T}(x^-|\bar{x}_u)\left(f(x)^\top f\left(x^-\right)\right)^2$$

$$+\eta_l^2\sum_{x,x^-\in\mathcal{X}}\mathbb{E}_{\bar{x}_u\sim\mathbb{P}}\mathcal{T}(x|\bar{x}_u)\mathbb{E}_{\bar{x}'_u\sim\mathbb{P}}\mathcal{T}(x^-|\bar{x}'_u)\left(f(x)^\top f\left(x^-\right)\right)^2$$

$$=\eta_u^2\sum_{i\in\mathcal{Y}_l}\sum_{j\in\mathcal{Y}_l}\underset{\substack{\bar{x}_l\sim\mathbb{P}_{l_i},\bar{x}'_l\sim\mathbb{P}_{l_j},\\x\sim\mathcal{T}(\cdot|\bar{x}_l),x^-\sim\mathcal{T}(\cdot|\bar{x}'_l)}}{\mathbb{E}}\left[\left(f(x)^\top f\left(x^-\right)\right)^2\right]$$

$$+2\eta_u\eta_l\sum_{i\in\mathcal{Y}_l}\underset{\substack{\bar{x}_l\sim\mathbb{P}_{l_i},\bar{x}_u\sim\mathbb{P},\\x\sim\mathcal{T}(\cdot|\bar{x}_l),x^-\sim\mathcal{T}(\cdot|\bar{x}_u)}}{\mathbb{E}}\left[\left(f(x)^\top f\left(x^-\right)\right)^2\right]$$

$$+\eta_l^2\underset{\substack{\bar{x}_u\sim\mathbb{P},\bar{x}'_u\sim\mathbb{P},\\x\sim\mathcal{T}(\cdot|\bar{x}_u),x^-\sim\mathcal{T}(\cdot|\bar{x}'_u)}}{\mathbb{E}}\left[\left(f(x)^\top f\left(x^-\right)\right)^2\right]$$

$$=\eta_u^2\mathcal{L}_3(f)+2\eta_u\eta_l\mathcal{L}_4(f)+\eta_l^2\mathcal{L}_5(f).$$

$\square$

## B  IMPACT OF SEMANTIC OOD DATA

In our main analysis in Section 4, we consider semantic OOD to be from a different domain. Alternatively, instances of semantic OOD data can come from the same domain as covariate OOD data. In this section, we provide a complete picture by contrasting these two cases.

**Setup.** In Figure 5, we illustrate two scenarios where the semantic OOD data has either a different or the same domain label as covariate OOD data. Other setups are the same as Sec. 4.3.

**Adjacency matrix.** The adjacency matrix for scenario (a) has been derived in Eq. 11. For the alternative scenario (b) where semantic OOD shares the same domain as the covariate OOD, we can derive the analytic form of adjacency matrix $A_1$.

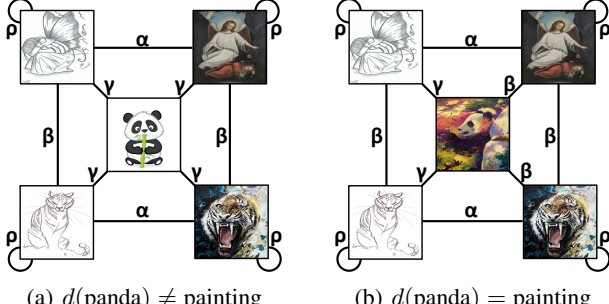

(a) $d(\text{panda}) \neq \text{painting}$   (b) $d(\text{panda}) = \text{painting}$

Figure 5: Illustration of 5 nodes graph and the augmentation probability defined by classes and domains. Figure (a) illustrates the scenario where semantic OOD data has a different domain from covariate OOD. Figure (b) depicts the case where semantic OOD and covariate OOD share the same domain.

$$\eta_u A_1^{(u)} = \begin{bmatrix} \rho^2+\beta^2+\alpha^2+2\gamma^2 & 2\rho\beta+\gamma^2+2\gamma\alpha & 2\rho\alpha+3\gamma\beta & 2\alpha\beta+\gamma\beta+2\gamma\rho & \alpha\beta+2\gamma(\beta+\rho) \\ 2\rho\beta+\gamma^2+2\gamma\alpha & \rho^2+\beta^2+\alpha^2+2\gamma^2 & 2\alpha\beta+\gamma\beta+2\gamma\rho & 2\rho\alpha+3\gamma\beta & \alpha\beta+2\gamma(\beta+\rho) \\ 2\rho\alpha+3\gamma\beta & 2\alpha\beta+\gamma\beta+2\gamma\rho & \rho^2+2\beta^2+\alpha^2+\gamma^2 & 2\rho\beta+\beta^2+2\gamma\alpha & 2\rho\beta+\beta^2+\gamma^2+\gamma\alpha \\ 2\alpha\beta+\gamma\beta+2\gamma\rho & 2\alpha\rho+3\gamma\beta & 2\rho\beta+\beta^2+2\gamma\alpha & \rho^2+2\beta^2+\alpha^2+\gamma^2 & 2\rho\beta+\beta^2+\gamma^2+\gamma\alpha \\ \alpha\beta+2\gamma(\beta+\rho) & \alpha\beta+2\gamma(\beta+\rho) & 2\rho\beta+\beta^2+\gamma^2+\gamma\alpha & 2\rho\beta+\beta^2+\gamma^2+\gamma\alpha & \rho^2+2\beta^2+2\gamma^2 \end{bmatrix} \tag{15}$$

$$A_1 = \frac{1}{C_1}(\eta_l A_1^{(l)} + \eta_u A_1^{(u)}) = \frac{1}{C_1}\left( \begin{bmatrix} \rho^2+\beta^2 & 2\rho\beta & \rho\alpha+\gamma\beta & \alpha\beta+\gamma\rho & \gamma(\beta+\rho) \\ 2\rho\beta & \rho^2+\beta^2 & \alpha\beta+\gamma\rho & \rho\alpha+\gamma\beta & \gamma(\beta+\rho) \\ \rho\alpha+\gamma\beta & \alpha\beta+\gamma\rho & \alpha^2+\gamma^2 & 2\gamma\alpha & \gamma(\gamma+\alpha) \\ \alpha\beta+\gamma\rho & \rho\alpha+\gamma\beta & 2\gamma\alpha & \alpha^2+\gamma^2 & \gamma(\gamma+\alpha) \\ \gamma(\beta+\rho) & \gamma(\beta+\rho) & \gamma(\gamma+\alpha) & \gamma(\gamma+\alpha) & 2\gamma^2 \end{bmatrix} + \eta_u A_1^{(u)} \right), \tag{16}$$

where $C_1$ is the normalization constant to ensure the summation of weights amounts to 1. Each row or column encodes connectivity associated with a specific sample, ordered by: angel sketch, tiger sketch, angel painting, tiger painting, and panda. We refer readers to the Appendix D.2 for the detailed derivation.

**Main analysis.** Following the same assumption in Sec. 4.3, we are primarily interested in analyzing the difference of the representation space derived from $A$ and $A_1$ and put analysis on the top-3 eigenvectors $\widehat{V}_1 \in \mathbb{R}^{5\times3}$.

**Theorem B.1.** *Denote $\alpha' = \frac{\alpha}{\rho}$ and $\beta' = \frac{\beta}{\rho}$ and assume $\eta_u = 5, \eta_l = 1$, we have:*

$$\widehat{V}_1 = \begin{bmatrix} \sqrt{2} & \sqrt{2} & 1 & 1 & 1 \\ a(\widehat{\lambda}_2) & a(\widehat{\lambda}_2) & b(\widehat{\lambda}_2) & b(\widehat{\lambda}_2) & 1 \\ c(\widehat{\lambda}_3) & -c(\widehat{\lambda}_3) & -1 & 1 & 0 \end{bmatrix}^{\top} \cdot R, \quad \mathcal{E}(f_1) = 0, \text{ if } \alpha > 0, \beta > 0. \tag{17}$$

*where $a(\lambda) = \frac{\sqrt{2}(1-6\beta'-\lambda)}{8\beta'}, b(\lambda) = \frac{4\beta'-1+\lambda}{4\beta'}, c(\lambda) = \frac{\sqrt{2}(1-3\alpha'-6\beta'-\lambda)}{3\alpha'}$. $R$ is a diagonal matrix that normalizes the eigenvectors to unit norm and $\widehat{\lambda}_2, \widehat{\lambda}_3$ are the 2nd and 3rd highest eigenvalues.*

**Interpretation.** When semantic OOD shares the same domain as covariate OOD, the OOD generalization error $\mathcal{E}(f_1)$ can be reduced to 0 as long as $\alpha$ and $\beta$ are positive. This generalization ability shows that semantic OOD and covariate OOD sharing the same domain could benefit OOD generalization. We empirically verify our theory in Section E.4.

**Theorem B.2.** *Denote $\alpha' = \frac{\alpha}{\rho}$ and $\beta' = \frac{\beta}{\rho}$ and assume $\eta_u = 5, \eta_l = 1$, we have:*

$$\mathcal{S}(f) - \mathcal{S}(f_1) \begin{cases} > 0 & \text{, if } \alpha', \beta' \in \text{black area in Figure 6 (b)}; \\ < 0 & \text{, if } \alpha', \beta' \in \text{white area in Figure 6 (b)}. \end{cases} \tag{18}$$

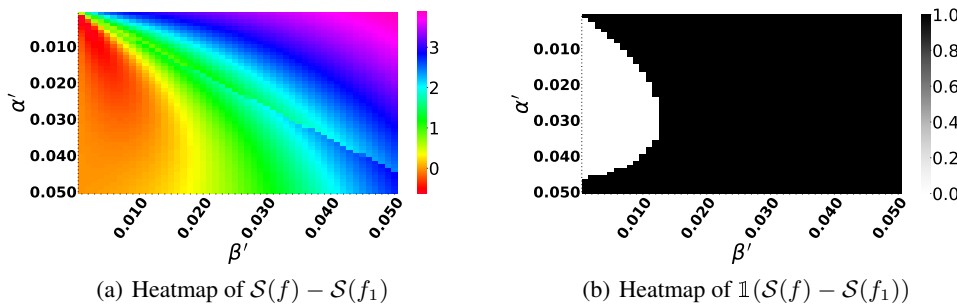

(a) Heatmap of $\mathcal{S}(f) - \mathcal{S}(f_1)$        (b) Heatmap of $\mathbb{1}(\mathcal{S}(f) - \mathcal{S}(f_1))$

Figure 6: Visualization of the separability difference between two cases defined in Figure 5 (a) and Figure 5 (b). Figure 6 (a) utilizes a heatmap to depict the distribution, while Figure 6 (a) uses the indicator function.

**Interpretation.** If $\alpha', \beta' \in$ black area in Figure 6 (b) and semantic OOD comes from a different domain, this would increase the separability between ID and semantic OOD, which benefits OOD detection. If $\alpha', \beta' \in$ white area in Figure 6 (b) and semantic OOD comes from a different domain, this would impair OOD detection.

## C    IMPACTS OF ID LABELS ON OOD GENERALIZATION AND DETECTION

Compared to spectral contrastive loss proposed by HaoChen et al. (2021), we utilize ID labels in the pre-training. In this section, we analyze the impacts of ID labels on the OOD generalization and detection performance.

Following the same assumption in Sec. 4.3, we are primarily interested in analyzing the difference of the representation space derived from $A$ and $A^{(u)}$ and put analysis on the top-3 eigenvectors $\widehat{V}^{(u)} \in \mathbb{R}^{5 \times 3}$. Detailed derivation can be found in the Appendix D.3.

**Theorem C.1.** *Assume $\eta_u = 5, \eta_l = 1$, we have:*

$$
\widehat{V}^{(u)} = \begin{cases} \frac{1}{2} \begin{bmatrix} 1 & 1 & 1 & 1 & 0 \\ 0 & 0 & 0 & 0 & 2 \\ -1 & 1 & -1 & 1 & 0 \end{bmatrix}^\top & , if\, \alpha > \beta; \\[18pt] \frac{1}{2} \begin{bmatrix} 1 & 1 & 1 & 1 & 0 \\ 0 & 0 & 0 & 0 & 2 \\ -1 & -1 & 1 & 1 & 0 \end{bmatrix}^\top & , if\, \alpha < \beta. \end{cases} , \mathcal{E}(f^{(u)}) = \begin{cases} 0 & , if\, \alpha > \beta; \\ 2 & , if\, \alpha < \beta. \end{cases} \quad (19)
$$

**Interpretation.** By comparing the eigenvectors $\widehat{V}$ in the supervised case (Theorem 4.1) and the eigenvectors $\widehat{V}^{(u)}$ in the self-supervised case, we find that adding ID label information transforms the performance condition from $\alpha = \beta$ to $\frac{9}{8}\alpha = \beta$. In particular, the discussion can be divided into two cases: (1) $\alpha > \beta$. (2) $\alpha < \beta$. In the first case when the connection between the class is stronger than the domain, the model could learn a perfect ID classifier based on features in the first two rows in $\widehat{V}^{(u)}$ and effectively generalize to the covariate-shifted domain (the third and fourth row in $\widehat{V}^{(u)}$), achieving perfect OOD generalization with $\mathcal{E}(f^{(u)}) = 0$. In the second case when the connection between the domain is stronger than the connection between the class, the embeddings of covariate-shifted OOD data are identical, resulting in high OOD generalization error.

**Theorem C.2.** *Assume $\eta_u = 5, \eta_l = 1$, we have:*

$$
\mathcal{S}(f) - \mathcal{S}(f^{(u)}) > 0,\, if\, \alpha > 0, \beta > 0 \quad (20)
$$

**Interpretation.** After incorporating ID label information, the separability between ID and semantic OOD in the learned embedding space increases as long as $\alpha$ and $\beta$ are positive. This suggests that ID label information indeed helps OOD detection. We empirically verify our theory in Section E.4.

## D  TECHNICAL DETAILS OF DERIVATION

### D.1  DETAILS FOR FIGURE 5 (A)

**Augmentation Transformation Probability**. Recall the augmentation transformation probability, which encodes the probability of augmenting an original image $\bar{x}$ to the augmented view $x$:

$$\mathcal{T}(x \mid \bar{x}) = \begin{cases} \rho & \text{if} & y(\bar{x}) = y(x), d(\bar{x}) = d(x); \\ \alpha & \text{if} & y(\bar{x}) = y(x), d(\bar{x}) \neq d(x); \\ \beta & \text{if} & y(\bar{x}) \neq y(x), d(\bar{x}) = d(x); \\ \gamma & \text{if} & y(\bar{x}) \neq y(x), d(\bar{x}) \neq d(x). \end{cases}$$

Thus, the augmentation matrix $\mathcal{T}$ of the toy example shown in Figure 5 (a) can be given by:

$$\mathcal{T} = \begin{bmatrix} \rho & \beta & \alpha & \gamma & \gamma \\ \beta & \rho & \gamma & \alpha & \gamma \\ \alpha & \gamma & \rho & \beta & \gamma \\ \gamma & \alpha & \beta & \rho & \gamma \\ \gamma & \gamma & \gamma & \gamma & \rho \end{bmatrix}$$

Each row or column encodes augmentation connectivity associated with a specific sample, ordered by: angel sketch, tiger sketch, angel painting, tiger painting, and panda.

**Details for $A^{(u)}$ and $A^{(l)}$**. Recall that the self-supervised connectivity is defined in Eq. 1. Since we have a 5-nodes graph, $A^{(u)}$ would be $\frac{1}{5}\mathcal{T}\mathcal{T}^\top$. If we assume $\eta_u = 5$, we can derive the closed-form self-supervised adjacency matrix:

$$\eta_u A^{(u)} = \begin{bmatrix} \rho^2 + \beta^2 + \alpha^2 + 2\gamma^2 & 2\rho\beta + \gamma^2 + 2\gamma\alpha & 2\rho\alpha + \gamma^2 + 2\gamma\beta & 2\alpha\beta + \gamma^2 + 2\gamma\rho & \gamma(\gamma + \alpha + \beta + 2\rho) \\ 2\rho\beta + \gamma^2 + 2\gamma\alpha & \rho^2 + \beta^2 + \alpha^2 + 2\gamma^2 & 2\alpha\beta + \gamma^2 + 2\gamma\rho & 2\rho\alpha + \gamma^2 + 2\gamma\beta & \gamma(\gamma + \alpha + \beta + 2\rho) \\ 2\rho\alpha + \gamma^2 + 2\gamma\beta & 2\alpha\beta + \gamma^2 + 2\gamma\rho & \rho^2 + \beta^2 + \alpha^2 + 2\gamma^2 & 2\rho\beta + \gamma^2 + 2\gamma\alpha & \gamma(\gamma + \alpha + \beta + 2\rho) \\ 2\alpha\beta + \gamma^2 + 2\gamma\rho & 2\rho\alpha + \gamma^2 + 2\gamma\beta & 2\rho\beta + \gamma^2 + 2\gamma\alpha & \rho^2 + \beta^2 + \alpha^2 + 2\gamma^2 & \gamma(\gamma + \alpha + \beta + 2\rho) \\ \gamma(\gamma + \alpha + \beta + 2\rho) & \gamma(\gamma + \alpha + \beta + 2\rho) & \gamma(\gamma + \alpha + \beta + 2\rho) & \gamma(\gamma + \alpha + \beta + 2\rho) & \rho^2 + 4\gamma^2 \end{bmatrix}$$

Then, according to the supervised connectivity defined in Eq. 2, we only compute ID-labeled data. Since we have two known classes and each class contains one sample, $A^{(l)} = \mathcal{T}_{:,1}\mathcal{T}_{:,1}^\top + \mathcal{T}_{:,2}\mathcal{T}_{:,2}^\top$. Then if we let $\eta_l = 1$, we can have the closed-form supervised adjacency matrix:

$$\eta_l A^{(l)} = \begin{bmatrix} \rho^2 + \beta^2 & 2\rho\beta & \rho\alpha + \gamma\beta & \alpha\beta + \gamma\rho & \gamma(\rho + \beta) \\ 2\rho\beta & \rho^2 + \beta^2 & \alpha\beta + \gamma\rho & \rho\alpha + \gamma\beta & \gamma(\rho + \beta) \\ \rho\alpha + \gamma\beta & \alpha\beta + \gamma\rho & \alpha^2 + \gamma^2 & 2\gamma\alpha & \gamma(\alpha + \gamma) \\ \alpha\beta + \gamma\rho & \rho\alpha + \gamma\beta & 2\gamma\alpha & \alpha^2 + \gamma^2 & \gamma(\alpha + \gamma) \\ \gamma(\rho + \beta) & \gamma(\rho + \beta) & \gamma(\alpha + \gamma) & \gamma(\alpha + \gamma) & 2\gamma^2 \end{bmatrix}$$

**Details of eigenvectors $\widehat{V}$.** We assume $\rho \gg \max(\alpha, \beta) \geq \min(\alpha, \beta) \gg \gamma \geq 0$, and denote $\alpha' = \frac{\alpha}{\rho}, \beta' = \frac{\beta}{\rho}$. $A$ can be approximately given by:

$$A \approx \widehat{A} = \frac{1}{\widehat{C}} \begin{bmatrix} 2 & 4\beta' & 3\alpha' & 0 & 0 \\ 4\beta' & 2 & 0 & 3\alpha' & 0 \\ 3\alpha' & 0 & 1 & 2\beta' & 0 \\ 0 & 3\alpha' & 2\beta' & 1 & 0 \\ 0 & 0 & 0 & 0 & 1 \end{bmatrix},$$

where $\widehat{C}$ is the normalization term and equals to $7 + 12\beta' + 12\alpha'$. The squares of the minimal term (e.g., $\frac{\alpha\beta}{\rho^2}, \frac{\alpha^2}{\rho^2}, \frac{\beta^2}{\rho^2}, \frac{\gamma}{\rho} = \frac{\gamma}{\alpha} \cdot \frac{\alpha}{\rho}, \frac{\alpha\gamma}{\rho^2}$, etc) are approximated to 0.

$$\widehat{D} = \frac{1}{\widehat{C}}\text{diag}[2 + 4\beta' + 3\alpha', 2 + 4\beta' + 3\alpha', 1 + 2\beta' + 3\alpha', 1 + 2\beta' + 3\alpha', 1]$$

$$\widehat{D^{-\frac{1}{2}}} = \sqrt{\widehat{C}}\text{diag}[\frac{1}{\sqrt{2}}(1 - \beta' - \frac{3}{4}\alpha'), \frac{1}{\sqrt{2}}(1 - \beta' - \frac{3}{4}\alpha'), 1 - \beta' - \frac{3}{2}\alpha', 1 - \beta' - \frac{3}{2}\alpha', 1]$$

$$D^{-\frac{1}{2}}AD^{-\frac{1}{2}} \approx \widehat{D^{-\frac{1}{2}}\widehat{A}D^{-\frac{1}{2}}} = \begin{bmatrix} 1 - 2\beta' - \frac{3}{2}\alpha' & 2\beta' & \frac{3}{\sqrt{2}}\alpha' & 0 & 0 \\ 2\beta' & 1 - 2\beta' - \frac{3}{2}\alpha' & 0 & \frac{3}{\sqrt{2}}\alpha' & 0 \\ \frac{3}{\sqrt{2}}\alpha' & 0 & 1 - 2\beta' - 3\alpha' & 2\beta' & 0 \\ 0 & \frac{3}{\sqrt{2}}\alpha' & 2\beta' & 1 - 2\beta' - 3\alpha' & 0 \\ 0 & 0 & 0 & 0 & 1 \end{bmatrix}$$

Let $\lambda_{1,...,5}$ and $v_{1,...,5}$ be the eigenvalues and their corresponding eigenvectors of $D^{-\frac{1}{2}}AD^{-\frac{1}{2}}$. Then the concrete form of $\lambda_{1,...,5}$ and $v_{1,...,5}$ can be approximately given by:

$$
\begin{aligned}
\widehat{v}_1 &= \tfrac{1}{\sqrt{6}}[\sqrt{2},\sqrt{2},1,1,0]^\top & \widehat{\lambda}_1 &= 1 \\
\widehat{v}_2 &= [0,0,0,0,1]^\top & \widehat{\lambda}_2 &= 1 \\
\widehat{v}_3 &= \tfrac{1}{\sqrt{6}}[-\sqrt{2},\sqrt{2},-1,1,0]^\top & \widehat{\lambda}_3 &= 1-4\beta' \\
\widehat{v}_4 &= \tfrac{1}{\sqrt{6}}[-1,-1,\sqrt{2},\sqrt{2},0]^\top & \widehat{\lambda}_4 &= 1-\tfrac{9}{2}\alpha' \\
\widehat{v}_5 &= \tfrac{1}{\sqrt{6}}[1,-1,-\sqrt{2},\sqrt{2},0]^\top & \widehat{\lambda}_5 &= 1-4\beta'-\tfrac{9}{2}\alpha'
\end{aligned}
$$

Since $\alpha', \beta' > 0$, we can always have $\widehat{\lambda}_1 = \widehat{\lambda}_2 > \widehat{\lambda}_3 > \widehat{\lambda}_5$ and $\widehat{\lambda}_1 = \widehat{\lambda}_2 > \widehat{\lambda}_4 > \widehat{\lambda}_5$. Then, we let $k = 3$ and $\widehat{V} \in \mathbb{R}^{5\times3}$ is given by:

$$
\widehat{V} = \begin{cases}
\begin{bmatrix}
\frac{1}{\sqrt{3}} & \frac{1}{\sqrt{3}} & \frac{1}{\sqrt{6}} & \frac{1}{\sqrt{6}} & 0 \\
0 & 0 & 0 & 0 & 1 \\
-\frac{1}{\sqrt{3}} & \frac{1}{\sqrt{3}} & -\frac{1}{\sqrt{6}} & \frac{1}{\sqrt{6}} & 0
\end{bmatrix}^\top & , \text{if } \frac{9}{8}\alpha' > \beta'; \\[4ex]
\begin{bmatrix}
\frac{1}{\sqrt{3}} & \frac{1}{\sqrt{3}} & \frac{1}{\sqrt{6}} & \frac{1}{\sqrt{6}} & 0 \\
0 & 0 & 0 & 0 & 1 \\
-\frac{1}{\sqrt{6}} & -\frac{1}{\sqrt{6}} & \frac{1}{\sqrt{3}} & \frac{1}{\sqrt{3}} & 0
\end{bmatrix}^\top & , \text{if } \frac{9}{8}\alpha' < \beta'.
\end{cases}
$$

**Details of linear probing and separability evaluation.** Recall that the closed-form embedding $Z = [D]^{-\frac{1}{2}}V_k\sqrt{\Sigma_k}$. Based on the derivation above, closed-form features for ID sample $Z_{\text{in}} \in \mathbb{R}^{2\times3}$ can be approximately given by:

$$
\widehat{Z}_{\text{in}} = \begin{cases}
\frac{(1-\beta'-0.75\alpha')\sqrt{\widehat{C}}}{\sqrt{6}}\begin{bmatrix} 1 & 0 & -\sqrt{1-4\beta'} \\ 1 & 0 & \sqrt{1-4\beta'} \end{bmatrix} & , \text{if } \frac{9}{8}\alpha' > \beta'. \\[3ex]
\frac{(1-\beta'-0.75\alpha')\sqrt{\widehat{C}}}{2\sqrt{3}}\begin{bmatrix} \sqrt{2} & 0 & -\sqrt{1-\frac{9}{2}\alpha'} \\ \sqrt{2} & 0 & -\sqrt{1-\frac{9}{2}\alpha'} \end{bmatrix} & , \text{if } \frac{9}{8}\alpha' < \beta'.
\end{cases}
$$

Based on the least error method, we can derive the weights of the linear classifier $M \in \mathbb{R}^{3\times2}$,

$$
\widehat{M} = (\widehat{Z}_{\text{in}}^\top\widehat{Z}_{\text{in}})^\dagger\widehat{Z}_{\text{in}}^T y_{\text{in}}
$$

where $(\cdot)^\dagger$ is the Moore-Penrose inverse and $y_{\text{in}}$ is the one-hot encoded ground truth class labels. So when $\frac{9}{8}\alpha > \beta$, the predicted probability $\widehat{y}_{\text{covariate}}$ can be given by:

$$
\widehat{y}_{\text{out}}^{\text{covariate}} = \widehat{Z}_{\text{out}}^{\text{covariate}} \cdot \widehat{M} = \frac{(1-\beta'-\frac{3}{2}\alpha')}{1-\beta'-\frac{3}{4}\alpha'} \cdot \mathcal{I}
$$

where $\mathcal{I} \in \mathbb{R}^{2\times2}$ is an identity matrix. We notice that when $\frac{9}{8}\alpha < \beta$, the closed-form features for ID samples are identical, indicating the impossibility of learning a clear boundary to classify classes angel and tiger. Eventually, we can derive the linear probing error:

$$
\mathcal{E}(f) = \begin{cases}
0 & , \text{if } \frac{9}{8}\alpha > \beta; \\[2ex]
2 & , \text{if } \frac{9}{8}\alpha < \beta.
\end{cases}
$$

The separability between ID data and semantic OOD data can be computed based on the closed-form embeddings $\widehat{Z}_{\text{in}}$ and $\widehat{Z}_{\text{out}}^{\text{semantic}}$:

$$
\widehat{Z}_{\text{out}}^{\text{semantic}} = \sqrt{\widehat{C}} \cdot [0,1,0]
$$

$$
\mathcal{S}(f) = \begin{cases}
(7+12\beta'+12\alpha')(\frac{1-2\beta'}{3}(1-\beta'-\frac{3}{4}\alpha')^2+1) & , \text{if } \frac{9}{8}\alpha > \beta; \\
(7+12\beta'+12\alpha')(\frac{2-3\alpha'}{8}(1-\beta'-\frac{3}{4}\alpha')^2+1) & , \text{if } \frac{9}{8}\alpha < \beta.
\end{cases}
$$

## D.2 DETAILS FOR FIGURE 5 (B)

**Augmentation Transformation Probability.** Illustrated in Figure 5 (b), when semantic OOD and covariate OOD share the same domain, the augmentation matrix can be slightly different from the previous case:

$$
\mathcal{T} = \begin{bmatrix}
\rho & \beta & \alpha & \gamma & \gamma \\
\beta & \rho & \gamma & \alpha & \gamma \\
\alpha & \gamma & \rho & \beta & \beta \\
\gamma & \alpha & \beta & \rho & \beta \\
\gamma & \gamma & \beta & \beta & \rho
\end{bmatrix}
$$

Each row or column represents augmentation connectivity of a specific sample, ordered by: angel sketch, tiger sketch, angel painting, tiger painting, and panda.

**Details for $A_1^{(u)}$ and $A_1^{(l)}$.** After the assumption $\eta_u = 5, \eta_l = 1$, we can have $\eta_u A_1^{(u)} = \mathcal{T}\mathcal{T}^\top$:

$$
\eta_u A_1^{(u)} = \begin{bmatrix}
\rho^2 + \beta^2 + \alpha^2 + 2\gamma^2 & 2\rho\beta + \gamma^2 + 2\gamma\alpha & 2\rho\alpha + 3\gamma\beta & 2\alpha\beta + \gamma\beta + 2\gamma\rho & \alpha\beta + 2\gamma(\beta + \rho) \\
2\rho\beta + \gamma^2 + 2\gamma\alpha & \rho^2 + \beta^2 + \alpha^2 + 2\gamma^2 & 2\alpha\beta + \gamma\beta + 2\gamma\rho & 2\rho\alpha + 3\gamma\beta & \alpha\beta + 2\gamma(\beta + \rho) \\
2\rho\alpha + 3\gamma\beta & 2\alpha\beta + \gamma\beta + 2\gamma\rho & \rho^2 + 2\beta^2 + \alpha^2 + \gamma^2 & 2\rho\beta + \beta^2 + 2\gamma\alpha & 2\rho\beta + \beta^2 + \gamma^2 + \gamma\alpha \\
2\alpha\beta + \gamma\beta + 2\gamma\rho & 2\alpha\rho + 3\gamma\beta & 2\rho\beta + \beta^2 + 2\gamma\alpha & \rho^2 + 2\beta^2 + \alpha^2 + \gamma^2 & 2\rho\beta + \beta^2 + \gamma^2 + \gamma\alpha \\
\alpha\beta + 2\gamma(\beta + \rho) & \alpha\beta + 2\gamma(\beta + \rho) & 2\rho\beta + \beta^2 + \gamma^2 + \gamma\alpha & 2\rho\beta + \beta^2 + \gamma^2 + \gamma\alpha & \rho^2 + 2\beta^2 + 2\gamma^2
\end{bmatrix}
$$

And the supervised adjacency matrix $A_1^{(l)} = \mathcal{T}_{:,1}\mathcal{T}_{:,1}^\top + \mathcal{T}_{:,2}\mathcal{T}_{:,2}^\top$ can be given by:

$$
\eta_l A_1^{(l)} = \begin{bmatrix}
\rho^2 + \beta^2 & 2\rho\beta & \rho\alpha + \gamma\beta & \alpha\beta + \gamma\rho & \gamma(\beta + \rho) \\
2\rho\beta & \rho^2 + \beta^2 & \alpha\beta + \gamma\rho & \rho\alpha + \gamma\beta & \gamma(\beta + \rho) \\
\rho\alpha + \gamma\beta & \alpha\beta + \gamma\rho & \alpha^2 + \gamma^2 & 2\gamma\alpha & \gamma(\gamma + \alpha) \\
\alpha\beta + \gamma\rho & \rho\alpha + \gamma\beta & 2\gamma\alpha & \alpha^2 + \gamma^2 & \gamma(\gamma + \alpha) \\
\gamma(\beta + \rho) & \gamma(\beta + \rho) & \gamma(\gamma + \alpha) & \gamma(\gamma + \alpha) & 2\gamma^2
\end{bmatrix}
$$

**Details for $\widehat{V}_1$.** Following the same assumption, the adjacency matrix can be approximately given by:

$$
A_1 \approx \widehat{A_1} = \frac{1}{\widehat{C_1}} \begin{bmatrix}
2 & 4\beta' & 3\alpha' & 0 & 0 \\
4\beta' & 2 & 0 & 3\alpha' & 0 \\
3\alpha' & 0 & 1 & 2\beta' & 2\beta' \\
0 & 3\alpha' & 2\beta' & 1 & 2\beta' \\
0 & 0 & 2\beta' & 2\beta' & 1
\end{bmatrix}
$$

$$
\widehat{D_1} = \frac{1}{\widehat{C_1}} \cdot \mathrm{diag}[2 + 4\beta' + 3\alpha', 2 + 4\beta' + 3\alpha', 1 + 4\beta' + 3\alpha', 1 + 4\beta' + 3\alpha', 1 + 4\beta']
$$

$$
\widehat{D_1^{-\frac{1}{2}}} = \sqrt{\widehat{C_1}} \cdot \mathrm{diag}[\frac{1}{\sqrt{2}}(1 - \beta' - \frac{3}{4}\alpha'), \frac{1}{\sqrt{2}}(1 - \beta' - \frac{3}{4}\alpha'), 1 - 2\beta' - \frac{3}{2}\alpha', 1 - 2\beta' - \frac{3}{2}\alpha', 1 - 2\beta']
$$

$$
D_1^{-\frac{1}{2}} A_1 D_1^{-\frac{1}{2}} \approx \widehat{D_1^{-\frac{1}{2}}} \widehat{A_1} \widehat{D_1^{-\frac{1}{2}}} = \begin{bmatrix}
1 - 2\beta' - \frac{3}{2}\alpha' & 2\beta' & \frac{3}{\sqrt{2}}\alpha' & 0 & 0 \\
2\beta' & 1 - 2\beta' - \frac{3}{2}\alpha' & 0 & \frac{3}{\sqrt{2}}\alpha' & 0 \\
\frac{3}{\sqrt{2}}\alpha' & 0 & 1 - 4\beta' - 3\alpha' & 2\beta' & 2\beta' \\
0 & \frac{3}{\sqrt{2}}\alpha' & 2\beta' & 1 - 4\beta' - 3\alpha' & 2\beta' \\
0 & 0 & 2\beta' & 2\beta' & 1 - 4\beta'
\end{bmatrix}
$$

where $\widehat{C_1}$ is the normalization term and $\widehat{C_1} = 7 + 20\beta' + 12\alpha'$. After eigendecomposition, we can derive ordered eigenvalues and their corresponding eigenvectors:

$$\widehat{v}_1 = \frac{1}{\sqrt{7}}[\sqrt{2}, \sqrt{2}, 1, 1, 1]^\top \qquad \widehat{\lambda}_1 = 1$$

$$\widehat{v}_2 = \frac{1}{\sqrt{2a(\widehat{\lambda}_2)^2 + 2b(\widehat{\lambda}_2)^2 + 1}}[a(\widehat{\lambda}_2), a(\widehat{\lambda}_2), b(\widehat{\lambda}_2), b(\widehat{\lambda}_2), 1]^\top \qquad \widehat{\lambda}_2 = 1 - 3b + \frac{\sqrt{3}\sqrt{(27a^2 - 40ab + 48b^2)} - 9a}{4}$$

$$\widehat{v}_3 = \frac{1}{\sqrt{2c(\widehat{\lambda}_3)^2 + 2}}[c(\widehat{\lambda}_3), -c(\widehat{\lambda}_3), -1, 1, 0]^\top \qquad \widehat{\lambda}_3 = 1 - 5b + \frac{\sqrt{81a^2 + 24ab + 16b^2} - 9a}{4}$$

$$\widehat{v}_4 = \frac{1}{\sqrt{2a(\widehat{\lambda}_4)^2 + 2b(\widehat{\lambda}_4)^2 + 1}}[a(\widehat{\lambda}_4), a(\widehat{\lambda}_4), b(\widehat{\lambda}_4), b(\widehat{\lambda}_4), 1]^\top \qquad \widehat{\lambda}_4 = 1 - 3b - \frac{\sqrt{3}\sqrt{(27a^2 - 40ab + 48b^2)} + 9a}{4}$$

$$\widehat{v}_5 = \frac{1}{\sqrt{2c(\widehat{\lambda}_5)^2 + 2}}[c(\widehat{\lambda}_5), -c(\widehat{\lambda}_5), -1, 1, 0]^\top, \qquad \widehat{\lambda}_5 = 1 - 5b - \frac{\sqrt{81a^2 + 24ab + 16b^2} + 9a}{4}$$

where $\widehat{\lambda}_1 > \widehat{\lambda}_2 > \widehat{\lambda}_3 > \widehat{\lambda}_4 > \widehat{\lambda}_5$ and $a(\lambda) = \frac{\sqrt{2}(1-6\beta'-\lambda)}{8\beta'}, b(\lambda) = \frac{4\beta'-1+\lambda}{4\beta'}, c(\lambda) = \frac{\sqrt{2}(1-3\alpha'-6\beta'-\lambda)}{3\alpha'}$. We can get closed-form eigenvectors:

$$
\widehat{V}_1 = \begin{bmatrix} \sqrt{2} & \sqrt{2} & 1 & 1 & 1 \\ a(\widehat{\lambda}_2) & a(\widehat{\lambda}_2) & b(\widehat{\lambda}_2) & b(\widehat{\lambda}_2) & 1 \\ c(\widehat{\lambda}_3) & -c(\widehat{\lambda}_3) & -1 & 1 & 0 \end{bmatrix}^\top \cdot \mathrm{diag}\left[\frac{1}{\sqrt{7}}, \frac{1}{\sqrt{2a(\widehat{\lambda}_2)^2 + 2b(\widehat{\lambda}_2)^2 + 1}}, \frac{1}{\sqrt{2c(\widehat{\lambda}_3)^2 + 2}}\right]
$$

**Details for linear probing and separability evaluation.** Following the same derivation, we can derive closed-form embedding for ID samples $\widehat{Z}_{\mathrm{in}} = \widehat{D_{\mathrm{in}}^{-\frac{1}{2}}} \widehat{V}_{\mathrm{in}} \sqrt{\widehat{\Sigma}_{\mathrm{in}}}$ and the linear layer weights $\widehat{M} = (\widehat{Z}_{\mathrm{in}}^\top \widehat{Z}_{\mathrm{in}})^\dagger \widehat{Z}_{\mathrm{in}}^T y_{\mathrm{in}}$. Eventually, we can derive the approximately predicted probability $\hat{y}_{\mathrm{out}}^{\mathrm{covariate}}$:

$$
\hat{y}_{\mathrm{out}}^{\mathrm{covariate}} = \begin{bmatrix} a_1 + b_1 & a_1 - b_1 \\ a_1 - b_1 & a_1 + b_1 \end{bmatrix}
$$

where $a_1, b_1 \in \mathbb{R}$ and $b_1 > 0$. This indicates that linear probing error $\mathcal{E}(f_1) = 0$ as long as $\alpha$ and $\beta$ are positive.

Having obtained closed-form representation $Z_{\mathrm{in}}$ and $Z_{\mathrm{out}}^{\mathrm{semantic}}$, we can compute separability $S(f_1)$ and then prove:

$$
\widehat{Z}_{\mathrm{in}} = \frac{(1 - \beta' - \frac{3}{4}\alpha')\sqrt{\widehat{C}_1}}{\sqrt{2}} \begin{bmatrix} \frac{\sqrt{2}}{\sqrt{7}} & \frac{a(\widehat{\lambda}_2)\sqrt{\widehat{\lambda}_2}}{\sqrt{2a(\widehat{\lambda}_2)^2 + 2b(\widehat{\lambda}_2)^2 + 1}} & -\frac{c(\widehat{\lambda}_3)\sqrt{\widehat{\lambda}_3}}{\sqrt{2c(\widehat{\lambda}_3)^2 + 2}} \\ \frac{\sqrt{2}}{\sqrt{7}} & \frac{a(\widehat{\lambda}_2)\sqrt{\widehat{\lambda}_2}}{\sqrt{2a(\widehat{\lambda}_2)^2 + 2b(\widehat{\lambda}_2)^2 + 1}} & \frac{c(\widehat{\lambda}_3)\sqrt{\widehat{\lambda}_3}}{\sqrt{2c(\widehat{\lambda}_3)^2 + 2}} \end{bmatrix}
$$

$$
\widehat{Z}_{\mathrm{out}}^{\mathrm{semantic}} = (1 - 2\beta')\sqrt{\widehat{C}_1}\left[\frac{1}{\sqrt{7}}, \frac{\sqrt{\widehat{\lambda}_2}}{\sqrt{2a(\widehat{\lambda}_2)^2 + 2b(\widehat{\lambda}_2)^2 + 1}}, 0\right]
$$

$$
\mathcal{S}(f) - \mathcal{S}(f_1) \begin{cases} > 0 & , \text{if } \alpha', \beta' \in \text{black area in Figure 6 (b)}; \\ < 0 & , \text{if } \alpha', \beta' \in \text{white area in Figure 6 (b)}. \end{cases}
$$

### D.3 CALCULATION DETAILS FOR SELF-SUPERVISED CASE

Our analysis for the self-supervised case is based on Figure 5 (a), the adjacency matrix is exactly the same as Eq. 10. After approximation, we can derive:

$$
A^{(u)} \approx \widehat{A}^{(u)} = \frac{1}{\widehat{C}^{(u)}} \begin{bmatrix} 1 & 2\beta' & 2\alpha' & 0 & 0 \\ 2\beta' & 1 & 0 & 2\alpha' & 0 \\ 2\alpha' & 0 & 1 & 2\beta' & 0 \\ 0 & 2\alpha' & 2\beta' & 1 & 0 \\ 0 & 0 & 0 & 0 & 1 \end{bmatrix}
$$

$$
\widehat{D^{(u)}}^{-\frac{1}{2}} = \sqrt{5 + 8\beta' + 8\alpha'} \cdot \mathrm{diag}[1 - \beta' - \alpha', 1 - \beta' - \alpha', 1 - \beta' - \alpha', 1 - \beta' - \alpha', 1]
$$

$$
\widehat{D^{(u)}}^{-\frac{1}{2}} \widehat{A^{(u)}} \widehat{D^{(u)}}^{-\frac{1}{2}} = \begin{bmatrix} 1 - 2\beta' - 2\alpha' & 2\beta' & 2\alpha' & 0 & 0 \\ 2\beta' & 1 - 2\beta' - 2\alpha' & 0 & 2\alpha' & 0 \\ 2\alpha' & 0 & 1 - 2\beta' - 2\alpha' & 2\beta' & 0 \\ 0 & 2\alpha' & 2\beta' & 1 - 2\beta' - 2\alpha' & 0 \\ 0 & 0 & 0 & 0 & 1 \end{bmatrix}
$$

$$
\begin{aligned}
\widehat{v}_1 &= \tfrac{1}{2}[1, 1, 1, 1, 0]^\top & \widehat{\lambda}_1 &= 1 \\
\widehat{v}_2 &= [0, 0, 0, 0, 1]^\top & \widehat{\lambda}_2 &= 1 \\
\widehat{v}_3 &= \tfrac{1}{2}[-1, 1, -1, 1, 0]^\top & \widehat{\lambda}_3 &= 1 - 4\beta' \\
\widehat{v}_4 &= \tfrac{1}{2}[-1, -1, 1, 1, 0]^\top & \widehat{\lambda}_4 &= 1 - 4\alpha' \\
\widehat{v}_5 &= \tfrac{1}{2}[1, -1, -1, 1, 0]^\top & \widehat{\lambda}_5 &= 1 - 4\alpha' - 4\beta'
\end{aligned}
$$

Following the same procedure presented above, we can prove Theorem C.1 and C.2.

# E  MORE EXPERIMENTS

## E.1  DATASET STATISTICS

We provide a detailed description of the datasets used in this work below:

**CIFAR-10** (Krizhevsky et al., 2009) contains $60,000$ color images with 10 classes. The training set has $50,000$ images and the test set has $10,000$ images.

**ImageNet-100** consists of a subset of 100 categories from ImageNet-1K (Deng et al., 2009). This dataset contains the following classes: n01498041, n01514859, n01582220, n01608432, n01616318, n01687978, n01776313, n01806567, n01833805, n01882714, n01910747, n01944390, n01985128, n02007558, n02071294, n02085620, n02114855, n02123045, n02128385, n02129165, n02129604, n02165456, n02190166, n02219486, n02226429, n02279972, n02317335, n02326432, n02342885, n02363005, n02391049, n02395406, n02403003, n02422699, n02442845, n02444819, n02480855, n02510455, n02640242, n02672831, n02687172, n02701002, n02730930, n02769748, n02782093, n02787622, n02793495, n02799071, n02802426, n02814860, n02840245, n02906734, n02948072, n02980441, n02999410, n03014705, n03028079, n03032252, n03125729, n03160309, n03179701, n03220513, n03249569, n03291819, n03384352, n03388043, n03450230, n03481172, n03594734, n03594945, n03627232, n03642806, n03649909, n03661043, n03676483, n03724870, n03733281, n03759954, n03761084, n03773504, n03804744, n03916031, n03938244, n04004767, n04026417, n04090263, n04133789, n04153751, n04296562, n04330267, n04371774, n04404412, n04465501, n04485082, n04507155, n04536866, n04579432, n04606251, n07714990, n07745940.

**CIFAR-10-C** is generated based on Hendrycks & Dietterich (2018), applying different corruptions on CIFAR-10 including gaussian noise, defocus blur, glass blur, impulse noise, shot noise, snow, and zoom blur.

**ImageNet-100-C** is generated with Gaussian noise added to ImageNet-100 dataset (Deng et al., 2009).

**SVHN** (Netzer et al., 2011) is a real-world image dataset obtained from house numbers in Google Street View images. This dataset $73,257$ samples for training, and $26,032$ samples for testing with 10 classes.

**Places365** (Zhou et al., 2017) contains scene photographs and diverse types of environments encountered in the world. The scene semantic categories consist of three macro-classes: Indoor, Nature, and Urban.

**LSUN-C** (Yu et al., 2015) and **LSUN-R** (Yu et al., 2015) are large-scale image datasets that are annotated using deep learning with humans in the loop. LSUN-C is a cropped version of LSUN and LSUN-R is a resized version of the LSUN dataset.

**Textures** (Cimpoi et al., 2014) refers to the Describable Textures Dataset, which contains a large dataset of visual attributes including patterns and textures. The subset we used has no overlap categories with the CIFAR dataset (Krizhevsky et al., 2009).

**iNaturalist** (Horn et al., 2018) is a challenging real-world dataset with iNaturalist species, captured in a wide variety of situations. It has 13 super-categories and 5,089 sub-categories. We use the subset from Huang & Li (2021) that contains 110 plant classes that no category overlaps with IMAGENET-1K (Deng et al., 2009).

**Office-Home** (Venkateswara et al., 2017) is a challenging dataset, which consists of 15500 images from 65 categories. It is made up of 4 domains: Artistic (Ar), Clip-Art (Cl), Product (Pr), and Real-World (Rw).

**Details of data split for OOD datasets.**  For datasets with standard train-test split (e.g., SVHN), we use the original test split for evaluation. For other OOD datasets (e.g., LSUN-C), we use $70\%$ of the data for creating the wild mixture training data as well as the mixture validation dataset. We use the remaining examples for test-time evaluation. For splitting training/validation, we use $30\%$ for validation and the remaining for training. During validation, we could only access unlabeled wild data and labeled clean ID data, which means hyper-parameters are chosen based on the performance of ID Acc. on the ID validation set (more in Section F).

| Model | Places365 $\mathbb{P}_{out}^{semantic}$, CIFAR-10-C $\mathbb{P}_{out}^{covariate}$ | | | | LSUN-R $\mathbb{P}_{out}^{semantic}$, CIFAR-10-C $\mathbb{P}_{out}^{covariate}$ | | | |
|---|---|---|---|---|---|---|---|---|
| | OOD Acc.↑ | ID Acc.↑ | FPR↓ | AUROC↑ | OOD Acc.↑ | ID Acc.↑ | FPR↓ | AUROC↑ |
| *OOD detection* | | | | | | | | |
| **MSP** | 75.05 | 94.84 | 57.40 | 84.49 | 75.05 | 94.84 | 52.15 | 91.37 |
| **ODIN** | 75.05 | 94.84 | 57.40 | 84.49 | 75.05 | 94.84 | 26.62 | 94.57 |
| **Energy** | 75.05 | 94.84 | 40.14 | 89.89 | 75.05 | 94.84 | 27.58 | 94.24 |
| **Mahalanobis** | 75.05 | 94.84 | 68.57 | 84.61 | 75.05 | 94.84 | 42.62 | 93.23 |
| **ViM** | 75.05 | 94.84 | **21.95** | **95.48** | 75.05 | 94.84 | 36.80 | 93.37 |
| **KNN** | 75.05 | 94.84 | 42.67 | 91.07 | 75.05 | 94.84 | 29.75 | 94.60 |
| **ASH** | 75.05 | 94.84 | 44.07 | 88.84 | 75.05 | 94.84 | 22.07 | 95.61 |
| *OOD generalization* | | | | | | | | |
| **IRM** | 77.92 | 90.85 | 53.79 | 88.15 | 77.92 | 90.85 | 34.50 | 94.54 |
| **GroupDRO** | 77.27 | **94.97** | 32.81 | 91.85 | 77.27 | 94.97 | 14.60 | 97.04 |
| **Mixup** | 79.17 | 93.30 | 58.24 | 75.70 | 79.17 | 93.30 | 32.73 | 88.86 |
| **VREx** | 76.90 | 91.35 | 56.13 | 87.45 | 76.90 | 91.35 | 44.20 | 92.55 |
| **EQRM** | 75.71 | 92.93 | 51.00 | 88.61 | 75.71 | 92.93 | 31.23 | 94.94 |
| **SharpDRO** | 79.03 | 94.91 | 34.64 | 91.96 | 79.03 | 94.91 | 13.27 | 97.44 |
| *Learning w. $\mathbb{P}_{wild}$* | | | | | | | | |
| **OE** | 35.98 | 94.75 | 27.02 | 94.57 | 46.89 | 94.07 | 0.70 | 99.78 |
| **Energy (w/ outlier)** | 19.86 | 90.55 | 23.89 | 93.60 | 32.91 | 93.01 | 0.27 | 99.94 |
| **Woods** | 54.58 | 94.88 | 30.48 | 93.28 | 78.75 | **95.01** | 0.60 | 99.87 |
| **Scone** | 85.21 | 94.59 | 37.56 | 90.90 | **80.31** | 94.97 | 0.87 | 99.79 |
| **SLW (Ours)** | $\mathbf{87.04}_{\pm0.3}$ | $93.40_{\pm0.3}$ | $40.97_{\pm1.1}$ | $91.82_{\pm0.0}$ | $79.38_{\pm0.8}$ | $92.44_{\pm0.1}$ | $\mathbf{0.06}_{\pm0.0}$ | $\mathbf{99.99}_{\pm0.0}$ |

Table 2: Additional results: comparison with competitive OOD generalization and OOD detection methods on CIFAR-10. To facilitate a fair comparison, we include results from Bai et al. (2023) and set $\pi_c = 0.5, \pi_s = 0.1$ by default for the mixture distribution $\mathbb{P}_{wild} := (1 - \pi_s - \pi_c)\mathbb{P}_{in} + \pi_s\mathbb{P}_{out}^{semantic} + \pi_c\mathbb{P}_{out}^{covariate}$. **Bold**=best. (*Since all the OOD detection methods use the same model trained with the CE loss on $\mathbb{P}_{in}$, they display the same ID and OOD accuracy on CIFAR-10-C.)

## E.2 RESULTS ON IMAGENET-100

In this section, we present results on the large-scale dataset ImageNet-100 to further demonstrate our empirical competitive performance. We employ ImageNet-100 as $\mathbb{P}_{in}$, ImageNet-100-C as $\mathbb{P}_{out}^{covariate}$, and iNaturalist (Horn et al., 2018) as $\mathbb{P}_{out}^{semantic}$. Similar to our CIFAR experiment, we divide the ImageNet-100 training set into 50% labeled as ID and 50% unlabeled. Then we mix unlabeled ImageNet-100, ImageNet-100-C, and iNaturalist to generate the wild dataset. We include results from Bai et al. (2023) and set $\pi_c = 0.5, \pi_s = 0.1$ for consistency. We pre-train the backbone ResNet-34 (He et al., 2016) with SLW and then use ID data to fine-tune the model. We set the pre-training epoch as 100, batch size as 512, and learning rate as 0.01. For fine-tuning, we set the learning rate to 0.01, batch size to 128, and train for 10 epochs. Empirical results in Table 3 indicate that our method effectively balances OOD generalization and detection while achieving strong performance in both aspects. While Wood (Katz-Samuels et al., 2022) displays strong OOD detection performance, the OOD generation performance (44.46%) is significantly worse than ours (72.58%). More detailed implementation can be found in Appendix F.

| Method | OOD Acc.↑ | ID Acc.↑ | FPR↓ | AUROC↑ |
|---|---|---|---|---|
| **Woods** | 44.46 | 86.49 | **10.50** | **98.22** |
| **Scone** | 65.34 | **87.64** | 27.13 | 95.66 |
| **SLW (Ours)** | **72.58** | 86.68 | 21.00 | 96.52 |

Table 3: Results on ImageNet-100. We employ ImageNet-100 as $\mathbb{P}_{in}$, ImageNet-100-C with Gaussian noise as $\mathbb{P}_{out}^{covariate}$, and iNaturalist as $\mathbb{P}_{out}^{semantic}$. **Bold**=Best.

## E.3 RESULTS ON OFFICE-HOME

In this section, we present empirical results on the Office-Home (Venkateswara et al., 2017), a dataset comprising 65 object classes distributed across 4 different domains: Artistic (Ar), Clipart (Cl), Product (Pr), and Real-World (Rw). Following Saito et al. (2018), we separate 65 object classes into the first 25 classes in alphabetic order as ID classes and the remainder of classes as semantic OOD classes. Subsequently, we construct the ID data from one domain (e.g., Ar) across 25 classes,

and the covariate OOD from another domain (e.g., Cl) to carry out the OOD generalization task (e.g., Ar → Cl). The semantic OOD data are from the remainder of classes, in the same domain as covariate OOD data. We consider the following wild data, where $\mathbb{P}_{\text{wild}} = \pi_c \mathbb{P}_{\text{out}}^{\text{covariate}} + \pi_s \mathbb{P}_{\text{out}}^{\text{semantic}}$ and $\pi_c + \pi_s = 1$. This setting is also known as open-set domain adaptation (Panareda Busto & Gall, 2017), which can be viewed as a special case of ours.

For a fair empirical comparison, we include results from Li et al. (2023), containing comprehensive baselines like STA (Liu et al., 2019), OSBP (Saito et al., 2018), DAOD (Fang et al., 2021), OSLPP (Wang et al., 2021b), ROS (Bucci et al., 2020), and Anna (Li et al., 2023). Following previous literature, we use OOD Acc. to denote the average class accuracy over known classes only in this section. We employ ResNet-50 (He et al., 2016) as the default backbone. As shown in Table 4, our approach strikes a balance between OOD generalization and detection, even outperforming the state-of-the-art method Anna in terms of FPR by 11.3% on average. This demonstrates the effectiveness of our method in handling the complex OOD scenarios present in the Office-Home dataset. More detailed implementation can be found in Appendix F.

| Method | Ar → Cl | | Ar → Pr | | Ar → Rw | | Cl → Ar | | Cl → Pr | | Cl → Rw | | Pr → Ar | |
|---|---|---|---|---|---|---|---|---|---|---|---|---|---|---|
| | OOD Acc.↑ | FPR↓ | OOD Acc.↑ | FPR↓ | OOD Acc.↑ | FPR↓ | OOD Acc.↑ | FPR↓ | OOD Acc.↑ | FPR↓ | OOD Acc.↑ | FPR↓ | OOD Acc.↑ | FPR↓ |
| STA$_{\text{sum}}$ | 50.8 | 36.6 | 68.7 | 40.3 | 81.1 | 49.5 | 53.0 | 36.1 | 61.4 | 36.5 | 69.8 | 36.8 | 55.4 | 26.3 |
| STA$_{\text{max}}$ | 46.0 | 27.7 | 68.0 | 51.6 | 78.6 | 39.6 | 51.4 | 35.0 | 61.8 | 40.9 | 67.0 | 33.3 | 54.2 | 27.6 |
| OSBP | 50.2 | 38.9 | 71.8 | 40.2 | 79.3 | 32.5 | 59.4 | 29.7 | 67.0 | 37.3 | 72.0 | 30.8 | 59.1 | 31.9 |
| DAOD | 72.6 | 48.2 | 55.3 | 42.1 | 78.2 | 37.4 | 59.1 | 38.3 | 70.8 | 47.4 | 77.8 | 43.0 | 71.3 | 49.5 |
| OSLPP | 55.9 | 32.9 | 72.5 | 26.9 | 80.1 | 30.6 | 49.6 | 21.0 | 61.6 | 26.7 | 67.2 | 26.1 | 54.6 | 23.8 |
| ROS | 50.6 | 25.9 | 68.4 | 29.7 | 75.8 | 22.8 | 53.6 | 34.5 | 59.8 | 28.4 | 65.3 | 27.8 | 57.3 | 35.7 |
| Anna | 61.4 | 21.3 | 68.3 | 20.1 | 74.1 | 20.3 | 58.0 | 26.9 | 64.2 | 26.4 | 66.9 | 19.8 | 63.0 | 29.7 |
| SLW (Ours) | 54.2 | 14.1 | 68.7 | 12.7 | 78.6 | 15.8 | 51.1 | 14.8 | 61.0 | 8.8 | 68.0 | 10.5 | 58.3 | 9.2 |

| Method | Pr → Cl | | Pr → Rw | | Rw → Ar | | Rw → Cl | | Rw → Pr | | Average | | | |
|---|---|---|---|---|---|---|---|---|---|---|---|---|---|---|
| | OOD Acc.↑ | FPR↓ | OOD Acc.↑ | FPR↓ | OOD Acc.↑ | FPR↓ | OOD Acc.↑ | FPR↓ | OOD Acc.↑ | FPR↓ | OOD Acc.↑ | FPR↓ | | |
| STA$_{\text{sum}}$ | 44.7 | 28.5 | 78.1 | 36.7 | 67.9 | 37.7 | 51.4 | 42.1 | 79.9 | 42.0 | 63.4 | 37.4 | | |
| STA$_{\text{max}}$ | 44.2 | 32.9 | 76.2 | 35.7 | 67.5 | 33.3 | 49.9 | 38.9 | 77.1 | 44.6 | 61.8 | 36.7 | | |
| OSBP | 44.5 | 33.7 | 76.2 | 28.3 | 66.1 | 32.7 | 48.0 | 37.0 | 76.3 | 31.4 | 64.1 | 33.7 | | |
| DAOD | 58.4 | 57.2 | 81.8 | 49.4 | 66.7 | 56.7 | 60.0 | 63.4 | 84.1 | 65.3 | 69.6 | 49.8 | | |
| OSLPP | 53.1 | 32.9 | 77.0 | 28.8 | 60.8 | 25.0 | 54.4 | 35.7 | 78.4 | 29.2 | 63.8 | 28.3 | | |
| ROS | 46.5 | 28.8 | 70.8 | 21.6 | 67.0 | 29.2 | 51.5 | 27.9 | 72.0 | 20.0 | 61.6 | 27.6 | | |
| Anna | 54.6 | 25.2 | 74.3 | 21.1 | 66.1 | 22.7 | 59.7 | 26.9 | 76.4 | 19.0 | 65.6 | 23.3 | | |
| SLW (Ours) | 48.1 | 13.4 | 76.9 | 8.00 | 64.8 | 9.5 | 56.1 | 11.8 | 80.9 | 14.5 | 63.9 | 12.0 | | |

Table 4: Results on Office-Home. **Bold**=Best.

## E.4 ABLATION STUDY

**Impacts of ID labels.** As shown in Table 5, we contrast performance by pre-training with and without ID labels. The wild data follows the same setting as our main paper, which is a composition of CIFAR-10, CIFAR-10-C, and one of the five semantic OOD datasets. By comparing OOD accuracy and FPR, we find that the use of ID labels during pre-training significantly improves both OOD generalization and OOD detection, which aligns with our theoretical analysis.

| $\mathbb{P}_{\text{out}}^{\text{semantic}}$ | ID labels | OOD Acc.↑ | ID Acc.↑ | FPR↓ | AUROC↑ |
|---|---|---|---|---|---|
| SVHN | ✗ | 62.02 | 80.26 | 20.64 | 96.44 |
| | ✓ | **86.62** | **93.10** | **0.13** | **99.98** |
| LSUN-C | ✗ | 67.59 | 83.35 | 57.70 | 88.83 |
| | ✓ | **85.88** | **92.61** | **1.76** | **99.75** |
| TEXTURES | ✗ | 64.47 | 76.78 | 75.66 | 78.32 |
| | ✓ | **81.40** | **92.50** | **12.05** | **98.25** |
| PLACES365 | ✗ | 70.76 | 81.48 | 66.40 | 83.15 |
| | ✓ | **87.04** | **93.40** | **40.97** | **91.82** |
| LSUN-R | ✗ | 63.09 | 74.25 | 40.50 | 90.24 |
| | ✓ | **79.68** | **92.44** | **0.06** | **99.99** |

Table 5: Impact of ID labels during pre-training. We employ CIFAR-10 as $\mathbb{P}_{\text{in}}$ and CIFAR-10-C with Gaussian noise as $\mathbb{P}_{\text{out}}^{\text{covariate}}$. **Bold**=Best.

**Impact of semantic OOD data.** Table 6 empirically verifies the theoretical analysis in Section B. We follow Cao et al. (2022) and separate classes in CIFAR-10 into 50% known and 50% unknown classes. To demonstrate the impacts of semantic OOD data on generalization, we simulate scenarios when semantic OOD shares the same or different domain as covariate OOD. Empirical results in Table 6 indicate that when semantic OOD shares the same domain as covariate OOD, it could significantly improve the performance of OOD generalization.

| Corruption Type of $\mathbb{P}^{\text{covariate}}_{\text{out}}$ | $\mathbb{P}^{\text{semantic}}_{\text{out}}$ | OOD Acc.↑ |
|---|---|---|
| Gaussian noise | SVHN | 85.48 |
| Gaussian noise | LSUN-C | 85.88 |
| Gaussian noise | Places365 | 83.28 |
| Gaussian noise | Textures | 86.84 |
| Gaussian noise | LSUN-R | 80.08 |
| Gaussian noise | Gaussian noise | **88.18** |

Table 6: The impact of semantic OOD data on generalization. Classes in CIFAR-10 are divided into 50% known and 50% unknown classes. The experiment in the last line uses known classes in CIFAR-10-C with Gaussian noise as $\mathbb{P}^{\text{covariate}}_{\text{out}}$ and novel classes in CIFAR-10-C with Gaussian noise as $\mathbb{P}^{\text{semantic}}_{\text{out}}$. **Bold**=best.

## F   IMPLEMENTATION DETAILS

**Training settings.** We conduct all the experiments in Pytorch, using NVIDIA GeForce RTX 2080Ti. We use SGD optimizer with weight decay 5e-4 and momentum 0.9 for all the experiments. In CIFAR-10 experiments, we pre-train Wide ResNet with SLW loss for 1000 epochs. The learning rate (lr) is 0.030, batch size (bs) is 512. Then we use ID-labeled data to fine-tune for 20 epochs with lr 0.005 and bs 512. In ImageNet-100 experiments, we train ImageNet pre-trained ResNet-34 with SLW loss for 100 epochs. The lr is 0.01, bs is 512. Then we fine-tune for 10 epochs with lr 0.01 and bs 128. In Office-Home experiments, we use ImageNet pre-trained ResNet-50 with lr 0.001 and bs 64. We use the same data augmentation strategies as SimSiam (Chen & He, 2021). We set K in KNN as 50 in CIFAR-10 experiments and 100 in ImageNet-100 experiments, which is consistent with Sun et al. (2022). And $\eta_u$ is selected within $\{1.00, 2.00\}$ and $\eta_l$ is within $\{0.02, 0.10, 0.50, 1.00\}$. In Office-Home experiments, we set K as 5, $\eta_u$ as 3, and $\eta_l$ within $\{0.01, 0.05\}$. $\eta_u, \eta_l$ are summarized in Table 7.

| ID/Covariate OOD | Semantics OOD | $\eta_l$ | $\eta_u$ |
|---|---|---|---|
| CIFAR-10/CIFAR-10-C | SVHN | 0.50 | 2.00 |
| CIFAR-10/CIFAR-10-C | LSUN-C | 0.50 | 2.00 |
| CIFAR-10/CIFAR-10-C | Textures | 0.50 | 1.00 |
| CIFAR-10/CIFAR-10-C | Places365 | 0.50 | 2.00 |
| CIFAR-10/CIFAR-10-C | LSUN-R | 0.10 | 2.00 |
| ImageNet-100/ImageNet-100-C | iNaturalist | 0.10 | 2.00 |
| Office-Home Ar/Cl, Pr, Rw | Cl, Pr, Rw | 0.01 | 3.00 |
| Office-Home Cl/Ar, Pr, Rw | Ar, Pr, Rw | 0.01 | 3.00 |
| Office-Home Pr/Ar, Cl, Rw | Ar, Cl, Rw | 0.05 | 3.00 |
| Office-Home Rw/Ar, Cl, Pr | Ar, Cl, Pr | 0.05 | 3.00 |

Table 7: Selection of hyper-parameters $\eta_l, \eta_u$

**Validation strategy.** For validation, we could only access to unlabeled mixture of validation wild data and clean validation ID data, which is rigorously adhered to Bai et al. (2023). Hyper-parameters are chosen based on the performance of ID Acc. on the ID validation set. We present the sweeping results in Table 8.

| $\eta_l$ | $\eta_u$ | ID Acc. (validation)↑ | ID Acc.↑ | OOD Acc.↑ | FPR↓ | AUROC↑ |
|------|------|------|------|------|------|------|
| 0.02 | 2.00 | 88.52 | 87.12 | 70.31 | 52.16 | 90.03 |
| 0.10 | 2.00 | 95.36 | 91.72 | 77.98 | 20.20 | 96.85 |
| 0.50 | 2.00 | 95.72 | 91.79 | 78.23 | 17.66 | 97.26 |
| 1.00 | 2.00 | 94.96 | 90.91 | 81.92 | 24.99 | 94.82 |
| 0.02 | 1.00 | 89.04 | 87.44 | 60.60 | 46.01 | 92.01 |
| 0.10 | 1.00 | 93.92 | 90.70 | 74.58 | 21.50 | 96.83 |
| 0.50 | 1.00 | **96.76** | 92.50 | 81.40 | 12.05 | 98.25 |
| 1.00 | 1.00 | 94.24 | 90.77 | 65.58 | 14.00 | 97.27 |

Table 8: Sensitivity analysis of hyper-parameters $\eta_l, \eta_u$. We employ CIFAR-10 as $\mathbb{P}_{in}$, CIFAR-10-C as $\mathbb{P}_{out}^{covariate}$, and Textures as $\mathbb{P}_{out}^{semantic}$. **Bold**=best.

