# OpenReview forum: "A Graph-Theoretic Framework for Joint OOD Generalization and Detection"
_ICLR.cc/2024/Conference — Submitted to ICLR 2024_

### Official Review · Reviewer_BLbH · 2023-10-31

**Soundness:** 3 good
**Presentation:** 3 good
**Contribution:** 3 good
**Rating:** 6
**Confidence:** 4

**Summary:**

The paper consider the OOD generalization and detection at the same time. The authors embed the distance between samples into a graph w.r.t. supervised and self-supervised signals. and extract principle components through the graph spectral analysis. The overall training method is applying contrastive loss on these decomposed components. The authors provide further analysis to illustrate the insights and conduct experiments to demonstrate the prominent performances.

**Strengths:**

1. The overall method is interesting and novel. Considering both OOD generalization and detection together and using graph to model the sample correlations instead of directly using distribution similarities is intriguing.
2. The organization of the paper and the demonstrations of the theoretical insights are fantastic.
3. The performances of the method is extraordinary good from both OOD generalization and detection aspects.

**Weaknesses:**

1. The authors state that all distribution types are encapsulated, but I think the paper does not consider concept shifts, where P(y|X) varies. Notably, this shift is not trivial, e.g., if feature $X_c$ is the cause of label $y$ but due to noises, $X_s$ has higher correlation with $y$ during training but not in testing, then the method may fail. In such a case, $X_s$ is a more prominent component during training, thus, it will be extracted by SVD decompositions and mislead model training including the contrastive training.
2. I did not find the OOD generalization theoretical guarantees and the required assumptions. According to my understanding, it is impossible to solve OOD generalization problem using purely observational data without any interventional equivalent information or assumptions. Can the authors clarify what are the assumptions and the guarantees?
3. The method may not be well motivated. I don't see why the authors choose a graph to model the sample correlations.

**Questions:**

1. May it be possible to analyze without constructing a graph and achieve the same training loss with SVD decompositions?

---

> ### Author Response · Authors · 2023-11-14
> **Response to Reviewer BLbH**
>
> We sincerely appreciate your positive feedback and constructive feedback! We address the questions below in detail.
>
> > **W1. Concept shifts**
>
> That's a very insightful comment! We focus on covariate or domain shift for the OOD generalization task, since it's one of the most studied forms of data set shift in popular benchmarks [1]. As you pointed out, we do recognize the existence of more challenging concept shifts, where the posterior distribution $p(y|x)$ changes while $p(x)$ remains the same. The adaptation in this setting can be fundamentally difficult without labeled target data. To estimate conditional distributions, one may need simultaneous observations of both variables.
>
> Admittedly, this setting would have been difficult for our framework to solve, since we focus on leveraging unlabeled data, and moreover, the unlabeled data can be mixed with other types of distributional shifts (e.g. semantic shifts). We concur with the reviewer, that the SVD decompositions on such joint distributions may not be able to accurately capture the posterior distribution in the target domain.
>
> With that being said, we do believe the problem posed here is very interesting and perhaps worth diving deeper in future work. We have revised our paper accordingly to reflect this.
>
> [1] Gulrajani, Ishaan et al. "In search of lost domain generalization." arXiv:2007.01434 (2020).
>
> > **W2. Theoretical guarantees and assumptions**
>
> We provide a theoretical guarantee for OOD generalization in Theorem 4.1, which bounds the linear probing error $\mathcal{E}(f)$ using the learned representations. As defined in Equation (7), the linear probing error measures the misclassification of linear head on covariate-shifted OOD data. We show that when the scaled connection between the class is stronger than the domain, the model could learn a perfect ID classifier and effectively generalize to the covariate-shifted domain, achieving perfect OOD generalization with linear probing error $\mathcal{E}(f)=0$.
>
> For the above theorem to hold, we assume the magnitude order of the probability of augmentation follows $\rho\gg \text{max}(\alpha,\beta)\ge\text{min}(\alpha,\beta)\gg\gamma\ge0$, where $\alpha$ indicates the augmentation probability when two samples share the same label but different domains, $\beta$ indicates the probability when two samples share different class labels but with the same domain, and $\gamma$ is the probability when two samples differ in both class and domain labels.
>
> Indeed, as you pointed out, generalization to arbitrary OOD can be impossible when the test distribution is unknown. Different from prior literature, our problem setting considers both ID data as well as unlabeled wild data (_which contains samples from the covariate-shifted domain_). Thus, the previous theory in the OOD-agnostic setting no longer applies to our case. We show that the generalization can provably reach low error when one can learn from the wild data, which provides new insight for the community.
>
> [2] Gilles Blanchard et al. Generalizing from several related classification tasks to a new unlabeled sample. In NeurIPS, 2011.
>
> [3] Krikamol Muandet et al. Domain generalization via invariant feature representation. In ICML, 2013
>
> > **W3. Motivation for choosing a graph to model the sample correlations.**
>
> Graph is a classic structure to model the connectivity among points, and reveal useful sub-structures. The sub-structures may correspond to images from different known classes, or OOD data with unknown classes. By performing spectral decomposition on such a graph, our driving motivation is to uncover meaningful structures for both OOD generalization and detection (e.g., covariate-shifted OOD data is close to the ID data, whereas semantic-shifted OOD data is distinguishable from ID data).
>
> Importantly, the graph allows us to theoretically understand how wild unlabeled data impacts OOD generalization and detection, through the lens of spectral graph theory. By way of spectral decomposition on the adjacency matrix $\widetilde{A}$, we can rigorously reason the OOD generalization and detection capability encoded in $\widetilde{A}$. As exemplified in Theorem 4.1 and Theorem 4.2, we analyze closed-form solutions for the OOD generalization and detection error. We believe our graph-based formulation provides a new angle to the community for the problems of OOD generalization and OOD detection, and may inspire more future work.
>
> > Q1. Without constructing a graph
>
> Our key insight is that our spectral contrastive loss (Equation 6) can be derived from a spectral decomposition of the graph adjacency matrix $\tilde A$. This loss effectively turns the eigendecomposition problem into a representation learning problem that can be optimized efficiently with neural networks. In other words, our loss function in Equation (6) allows bypassing the graph construction and eigendecomposition, and can be optimized as a form of contrastive learning objective.

---

> > ### Comment · Reviewer_BLbH · 2023-11-20
> >
> > Thank you for your response. I believe the overall paper quality is satisfactory and leave my score as is. However, I urge the authors to add a comprehensive limitation section in the appendix including a classification of what distribution shifts are covered and what are not.

---

> > > ### Author Response · Authors · 2023-11-20
> > > **thank you!**
> > >
> > > Dear reviewer BLbH,
> > >
> > > We appreciate your suggestion and will make sure to include such a discussion in the paper. Once again thank you for the positive feedback, which means a lot to us.
> > >
> > > Authors

---

### Official Review · Reviewer_uUKR · 2023-10-31

**Soundness:** 3 good
**Presentation:** 2 fair
**Contribution:** 3 good
**Rating:** 3
**Confidence:** 3

**Summary:**

The paper proposes to handle out-of-distribution generalization and detection tasks in a unified method. To this end, the authors propose a graph-based approach by constructing a self-supervised graph adjacency and a supervised graph adjacency based on the augmentation views of the data. Theoretical analysis is provided to justify the design from the perspective of feature decomposition and separability evaluation. Experiments on common benchmarks validate the proposed model.

**Strengths:**

1. The idea seems novel to my knowledge, and the model is reasonable and sound

2. The intuition behind the method is clearly described and the theoretical analysis well justified the model

3. The experimental results look promising and the improvements are significant.

**Weaknesses:**

1. The major concern for me on this work is the potential overclaim. The paper title describes the method as a "graph-theoretic framework", but instead, the method is purely based on some heuristic ways for constructing a graph from the data. Also, the theoretical analysis has weak connection with graph theory, and is based on linear algebra. Furthermore, the existing experiments fail to show how the method can act as a general "framework" where different models can be applied. From my personal view, I would recognize the method as a graph-based model instead of a "graph-theoretic framework".

2. The second concern is the limited evaluation in comparison with the broad claim in the title and introduction. For out-of-distribution generalization, there are diverse kinds of distirbution shifts in data, and the current work only studies one particular type, covariate shift. It limits the scope of this paper against the claim of "unifying out-of-distribution generalization and detection". Also, the selection for baselines is not that convincing. The baselines for out-of-distribution generalization tasks are published several years ago, and there are plenty of SOTA models that are missing. For OOD detection, the scores appeared in the paper are different from those reported in the paper of baselines. Is this due to that the authors use different protocols?

3. Some of the descriptions are misleading. E.g., the self-supervised graph adjacency and supervised graph adjacency. Also the section tiltle for 3.1 "Graph-theoretic formulation" is misguided. What does the graph theory refer to and how is the problem of OOD learning formulated as a graph problem?

**Questions:**

See the weakness section

---

> ### Author Response · Authors · 2023-11-14
> **Response to Reviewer uUKR**
>
> We thank the reviewer for recognizing the novelty and soundness of our approach. We are encouraged that you appreciate the theoretical justification and our experimental results. Below we address your comments in detail.
>
> > **W1. Clarification on terminology "graph-theoretic framework"**
>
> We thank you for raising this concern. The central component of our framework is to perform spectral decomposition to the population graph to construct principled embeddings for OOD generalization and OOD detection (see Section 3.2). Such a decomposition is fundamentally related to **spectral graph theory** [1,2]. Spectral graph theory is a classic research field, concerning the study of graph partitioning through analyzing the eigenspace of the adjacency matrix. We provided an extensive discussion on spectral graph theory in the related work section (page 9), along with its connection to modern machine learning.
>
> We would also like to point out that the terminology of _graph-theoretic_ framework is adopted from the recent pioneering work of HaoChen et al.[3], which provided a theoretical analysis of unsupervised learning. Different from previous literature, our work focuses on the joint problem of OOD generalization and detection, which has fundamentally different data setup and learning goals (cf. Section 2). In particular, we are dealing with unlabeled data with heterogeneous mixture distribution, which is more general and challenging than previous works. We are interested in leveraging labeled data to classify some unlabeled data correctly into the known categories while rejecting the remainder of unlabeled data from new categories. Accordingly, we derive a novel theoretical analysis uniquely tailored to our problem setting, as shown in Section 4.
>
> With that being said, we believe the suggested terminology of _graph-based framework_ is also applicable, and will not affect the essence of our contributions. We have accordingly revised our manuscript (changes marked in blue color). Thank you again for pointing this out!
>
>
>
> [1] Fan RK Chung and Fan Chung Graham. Spectral graph theory. Number 92. American Mathematical Soc., 1997.
>
> [2] Jeff Cheeger. A lower bound for the smallest eigenvalue of the laplacian. In Proceedings of the Princeton conference in honor of Professor S. Bochner, pages 195–199, 1969.
>
> [3] Jeff Z. HaoChen, Colin Wei, Adrien Gaidon, and Tengyu Ma. Provable guarantees for self-supervised deep learning with spectral contrastive loss. In NeurIPS, pp. 5000–5011, 2021.
>
>
> >**W2. Additional evaluations, baselines**
>
> Literature in OOD generalization commonly considers covariate shift and domain shift. For domain shift, we have included additional evaluations on Office-Home in **Appendix E.3**, where the competitiveness of our method holds.
>
> For baselines, we follow closely the latest work SCONE [4], which considers the identical problem setting as ours. In general, methods that have access to wild data are more competitive than standard OOD generalization (which only has access to in-distribution data only). We believe our results and comparisons are meaningful since we already included the SOTA method SCONE.
>
> To make our comparison with OOD generalization baselines more convincing, we provide results of EQRM [5] and the latest SOTA baseline from CVPR'23 called SharpDRO [6]. The results of employing SVHN as semantic OOD dataset are shown in the table below (CIFAR-10 as ID, CIFAR-10-C as covariate-shift OOD). Compared to the SOTA baseline tailored for OOD generalization, our method can improve OOD Acc. by 7.59%.
> | Method       | OOD Acc. $\uparrow$  | ID Acc. $\uparrow$  | FPR $\downarrow$     | AUROC $\uparrow$    |
> | ------------ | --------- | --------- | -------- | --------- |
> | EQRM [5]     | 75.71     | 92.93     | 51.86    | 90.92     |
> | SharpDRO [6] | 79.03     | 94.91     | 21.24    | 96.14     |
> | SLW (Ours)   | 86.62±0.3 | 93.10±0.1 | 0.13±0.0 | 99.98±0.0 |
>
>
> The numbers for OOD detection baselines are consistent with Table 3 in Bai et al. Strictly following [4], we used the Wide ResNet with 40 layers and a widen factor of 2 to conduct our experiments. The results may differ from those reported in the original baseline papers due to different network architectures and pre-training checkpoints.
>
>
> [4] Bai, Haoyue et al. Feed two birds with one scone: Exploiting wild data for both out-of-distribution generalization and detection. In ICML, 2023.
>
> [5] Eastwood, Cian et al. Probable domain generalization via quantile risk minimization. In NeurIPS, 2022.
>
> [6] Huang, Zhuo et al. Robust Generalization against Photon-Limited Corruptions via Worst-Case Sharpness Minimization. In CVPR 2023.
>
> > **W3. Terminology in Section 3.1**
>
> We do agree with the reviewer that the title of Section 3.1 should be revised, since spectral graph theory only becomes relevant in subsequent sections. As suggested, we have changed it to simply "Graph Formulation". Thanks again for the suggestion!

---

### Official Review · Reviewer_WvgX · 2023-10-31

**Soundness:** 3 good
**Presentation:** 3 good
**Contribution:** 3 good
**Rating:** 6
**Confidence:** 4

**Summary:**

This paper introduces an innovative graph-theoretical framework that addresses the dual challenges of OOD generalization and detection. The foundational approach in the paper involves constructing a graph wherein nodes represent image data, and edges are established to connect similar data points. The definition of edges, influenced by both labeled and unlabeled data, lays the groundwork for examining OOD generalization and detection through a spectral lens.

The paper further contributes by proposing a spectral contrastive loss, which facilitates concurrent learning from labeled in-distribution (ID) data and unlabeled wild data. This loss function serves as a key component of the framework, contributing to the overall effectiveness of the method.

Moreover, the paper substantiates its claims through a series of well-conducted experiments that illustrate the capabilities of the framework in the context of OOD generalization and detection. These experiments not only validate the proposed approach but also provide valuable insights into its practical utility.

**Strengths:**

The author introduces a novel framework aimed at addressing both the challenges of out-of-distribution (OOD) generalization and OOD detection simultaneously. This is achieved through the spectral decomposition of a graph that encompasses in-distribution (ID) data, covariate-shift OOD data, and semantic-shift OOD data.

The paper not only presents this innovative approach but also offers valuable theoretical insights into the learned representations. It is worth noting that the paper claims that the closed-form solution for the representation is equivalent to conducting spectral decomposition of the adjacency matrix.

Furthermore, the experimental results, conducted on various datasets, showcase the impressive performance of the Spectral Learning with Wild data (SLW) framework. The evidence provided through these experiments underscores the framework's competitiveness and potential to make significant contributions to the field of OOD generalization and detection.

The paper's combination of theoretical analysis and empirical validation, along with its innovative approach, suggests that it has the potential to bring valuable advancements to the understanding and application of spectral decomposition in addressing OOD challenges. However, some additional clarifications and refinements may be needed to fully grasp the scope and implications of the proposed framework

**Weaknesses:**

•	Confusion Between Augmented Graph and Image: The paper appears to introduce both augmented graphs and augmented images, but the relationship and distinction between these concepts are not well-defined. The authors should provide a more comprehensive explanation of how these two augmentation techniques are related and used in conjunction within the paper. A clear rationale for why both augmented graphs and images are necessary should be provided to justify their inclusion.

•	Model complexity: Constructing the adjacency matrix, which encodes image connectivity, presents challenges in terms of both its creation and computational demands. The difficulty associated with building the adjacency matrix, along with the potential challenges of computation in real-world data, warrants careful consideration.

•	The paper introduces the term \frac{9}{8}\alpha without a clear derivation or explanation of its significance. It is crucial to provide a detailed and step-by-step derivation of this term to enhance the paper's mathematical rigor.

•	Typos, for example, there is not x^+ in equation (1)

**Questions:**

The paper should engage in a meaningful discussion on the potential for extending the framework to a more diverse dataset. This discussion should touch upon the adaptability and robustness of the proposed methods, models, or techniques.

---

> ### Author Response · Authors · 2023-11-14
> **Response to Reviewer WvgX**
>
> We sincerely appreciate your positive feedback and insightful comments! We address the questions below in detail.
>
> > **W1. Confusion Between Augmented Graph and Image**
>
> Thank you for pointing this out! In Section 3.1, we introduce the definition of _graph_ $G(\mathcal{X},w)$, where the vertex set $\mathcal{X}$ consists of all _augmented images_.
>
> For any two augmented images $x$ and $x'\in \mathcal{X}$, we define the weight $w_{xx'}$ based on Equation (2).
>
> $$\begin{align}
> \begin{split}
> w_{x x^{\prime}} = \eta_{u} w^{(u)}\_{x x^{\prime}} + \eta\_{l} w^{(l)}_{x x^{\prime}}
> \end{split}
> \end{align}$$
> where
> $$\begin{align}
> w^{(u)}\_{x x^{\prime}} & \triangleq \mathbb{E}\_{\bar{x} \sim {\mathbb{P}}}  \mathcal{T}(x| \bar{x}) \mathcal{T}(x'| \bar{x}) \\\\
> w^{(l)}\_{x x^{\prime}} & \triangleq \sum\_{i \in \mathcal{Y}\_l}\mathbb{E}\_{\bar{x}\_{l} \sim {\mathbb{P}\_{l\_i}}} \mathbb{E}\_{\bar{x}'\_{l} \sim {\mathbb{P}\_{l\_i}}} \mathcal{T}(x | \bar{x}\_{l}) \mathcal{T}\left(x' | \bar{x}'\_{l}\right).
> \end{align}$$
>
> Here, $\mathcal{T}(x|\bar{x})$ denotes the probability of $x$ being augmented from $\bar{x}$. In other words, we can derive the graph based on the augmentation transformation and its probability. The relative magnitude of $w_{xx'}$ intuitively captures the closeness between $x$ and $x'$ with respect to the augmentation transformation. For most of the unrelated $x$ and $x'$, the value $w_{xx'}$ will be significantly smaller than the average value. For example, when $x$ and $x'$ are random croppings of a cat and a dog respectively, $w_{xx'}$ will be essentially zero because no natural data can be augmented into both $x$ and $x'$.
>
> We understand that confusion may arise from the wording of "augmentation graph" on page 6, which simply encodes the augmentation probability between any two images $x$ and $x'$. We have revised the draft accordingly and changed it to "augmentation transformation probability", which hopefully avoids the confusion.
>
>
>
>
> > **W2. Model complexity**
>
> Indeed, as you concur, directly performing eigendecomposition on the graph may be computationally intractable for real-world data with many images. That's precisely the reason for proposing the spectral loss (Section 3.2), which turns the eigendecomposition problem into a representation learning problem that can be optimized efficiently with neural networks. In particular, we parameterize the rows of the eigenvector matrix as a neural net function and assume embeddings can be represented by $f(x)$ for some $f \in \mathcal{F}$, where $\mathcal{F}$ is the hypothesis class containing neural networks. This provides us with the convenience of utilizing the power of neural networks and learning on a large amount of data. In other words, our loss function in Equation (6) allows bypassing the graph construction and eigendecomposition, and can be optimized efficiently as a form of contrastive learning objective (by pulling closer the positive pairs and pushing apart the negative pairs).
>
> Importantly, our learning objective allows us to draw a theoretical equivalence between learned representations and the top-$k$ singular vectors of the normalized adjacency matrix $\tilde{A}$. Such equivalence facilitates theoretical understanding of the OOD generalization and OOD detection capability encoded in $\tilde{A}$, while enjoying the benefits of being end-to-end trainable.
>
> Empirically, we have demonstrated that our method can be applicable to real-world datasets including ImageNet (Appendix E.2), without the computation issue.
>
>
> > **W3. Derivation**
>
> For your reference, the full derivation can be found in **Appendix D**, pages 20-21. In short, the coefficient comes from the closed-form derivation of the top singular values.
>
> > **W4. Typos**
>
> Yes, $x^+$ should be replaced with $x'$. This has been fixed in our new draft. Thanks for the careful read!
>
> > **Q1. Experiments on diverse datasets**
>
> We agree that such an extension would be meaningful and important to support the broad applicability of our framework. Apart from CIFAR-10, we also provide large-scale results on the ImageNet dataset in **Appendix E.2** and additional results on the OfficeHome dataset in **Appendix E.3**.

---

### Author Response · Authors · 2023-11-14
**Response summary**

We thank all the reviewers for their time and commitment to providing valuable feedback and suggestions on our work. We are encouraged that ALL reviewers find our paper _novel_ (WvgX, uUKR, BLbH), that our methodology and theoretical insights are _interesting, sound, and valuable_ (WvgX, uUKR, BLbH), and that our results are _impressive and significant_ (WvgX, uUKR, BLbH). We appreciate that reviewers acknowledge our _clear organization and presentation_ (uUKR, BLbH).

We have responded to all comments and questions from each reviewer in detail below. We have also modified the manuscript in response to the reviewer's suggestions. The main changes are highlighted in blue, which include:

+ Replaced "graph-theoretic" with "graph-based" throughout the paper.
+ Replaced "augmentation graph" with "augmentation transformation probability" (Section 4.3)
+ Added the latest SOTA baseline for OOD generalization (Section 5)
+ Fixed typos (Section 3.1)

We believe these changes have helped strengthen our manuscript. Once again, we express our gratitude for your thoughtful and thorough evaluations.

Sincerely,

authors

---

### Meta-Review · Area_Chair_Kr1o · 2023-12-06

**Metareview:**

This paper addresses both out-of-domain generalization and detection by proposing a graph-theoretical frameworks dealing with these two problems. A method based on spectral learning is proposed accompanied with a theoretical analysis and an empirical evaluation.


On the positive side, the reviews have identified that the paper presents a novel framework for OOD generalization and detection with graphs, the method appears to be sound, the paper provides both theoretical and experimental contributions, the experiments are convincing, the paper is well-organized.
On the negative side, some elements require more description: links between augmented graph and images, complexity of the model, graph theory considered, the type of shifts considered, motivation of the method; experiments should consider more diverse datasets and baselines.

Authors provided detailed response in the rebuttal and a new revision of the paper that include the modification of some wording, correction of typos and the addition of one SOTA baseline.
During discussion, the question of the shifts covered was discussed and the fact that the paper mail focuses on covariate shifts and not concept shift was seen as a limitation. The revision appears globally limited with mainly some local modifications of wording with addition of few baseline. The paper would benefit of more thorough revision to have a better presentation and positioning with respect to the state of the art notably regarding the use of spectral graph theory, this would allow slo to offer a more comprehensive presentation of the theoretical analysis and notably its novelty.


I propose then rejection.
I encourage nevertheless the authors to take into account these remarks to improve the paper for other venues.

**Justification For Why Not Higher Score:**

Some issues about a potential limited set of shifts considered (e.g. no concept shifts) and lack of positioning with respect to SOTA on spectral graph theory were among the negative points considered for rejection.

**Justification For Why Not Lower Score:**

N/A

---

### Decision · Program_Chairs · 2024-01-16

Reject